# Genomic surveillance reveals a dengue 2 virus epidemic lineage with a marked decrease in sensitivity to Mosnodenvir

Hawa Sophia Bouzidi[1,4], Selin Sen [1,4], Géraldine Piorkowski[1], Laura Pezzi[1,2], Nazli Ayhan[1,2], Albin Fontaine [1,3], Thomas Canivez[1,2], Manon Geulen[1,2], Rayane Amaral[1], Gilda Grard[1,2], Guillaume André Durand[1,2], Xavier de Lamballerie [1,2], Franck Touret [1,5] ✉ & Raphaëlle Klitting [1,2,5] ✉

Dengue fever is the most important arbovirosis for public health, with more than 5 million cases worldwide in 2023. Mosnodenvir is the first anti-dengue compound with very high preclinical pan-serotype activity, currently undergoing phase 2 clinical evaluation. Here, by analyzing dengue virus (DENV) genomes from the 2023-2024 epidemic in the French Caribbean Islands, we show that they all exhibit mutation NS4B:V91A, previously associated with a marked decrease in sensitivity to mosnodenvir in vitro. Using antiviral activity tests on four clinical and reverse-genetic strains, we confirm a marked decrease in mosnodenvir sensitivity for DENV-2 ( > 1000 fold). Finally, combining phylogenetic analysis and experimental testing for resistance, we find that virus lineages with low sensitivity to mosnodenvir due to the V91A mutation likely emerged multiple times over the last 30 years in DENV-2 and DENV-3. These results call for increased genomic surveillance, in particular to track lineages with resistance mutations. These efforts should allow to better assess the activity profile of DENV treatments in development against circulating strains.

Dengue is the most prominent arbovirosis in terms of public health. The number of reported dengue cases rose 10-fold between 2000 and 2019[1] and around half of the world population is estimated to be currently at risk of contracting the disease[2]. Dengue is caused by dengue virus (DENV), a mosquito-borne virus belonging to the *Orthoflavivirus* genus and including 4 different serotypes (1-4). The infection is primarily transmitted through the bite of mosquitoes from the species *Aedes aegypti*, widely distributed across tropical and subtropical latitudes, where the disease is considered endemic[3]. With the expansion in the range of a secondary vector *Ae. albopictus*[4], autochthonous cases of dengue are increasingly reported beyond the subtropics including in Europe, where more than 100 local cases were identified in 2023[1].

The World Health Organization (WHO) reported exceptional numbers for dengue in 2023, with more than 5 million cases and over 5000 dengue-related deaths, distributed across more than 80 countries and territories[2]. Among these territories, the French Caribbean Islands have been experiencing intense DENV circulation due to DENV-2 genotype II (also called Cosmopolitan[5]). The islands of Guadeloupe, Martinique, Saint Martin and Saint Barthélemy all entered a state of epidemic during the second half of 2023, with more than 30,000 cases as of March 2024[6]. A case count that amounts to a high incidence − above 38 cases per 1000 inhabitants given the islands latest population counts[7–9]. In comparison, the incidence reported for last year in Brazil was 1 359 cases per 100 000

[1]Unité des Virus Émergents (Aix-Marseille Université, Università di Corsica, IRD 190, Inserm 1207, IRBA), Marseille, France. [2]Centre National de Référence des Arbovirus, Inserm-IRBA, Marseille, France. [3]Institut de Recherche Biomédicale des Armées (IRBA), Unité de virologie, Marseille, France. [4]These authors contributed equally: Hawa Sophia Bouzidi, Selin Sen. [5]These authors jointly supervised this work: Franck Touret, Raphaëlle Klitting. ✉e-mail: franck.touret@ird.fr; raphaelle.klitting@inserm.fr

population[1] that is, ~13 cases par 1 000 inhabitants. This ongoing epidemic appears to be driven by a single main lineage that circulates in the four islands and has also become established in French Guiana –another French overseas territory located in South America that shares strong ties with the Caribbean Islands[10].

Vector control has long been the first-line intervention to prevent dengue but in recent years a first vaccine, DENGVAXIA[11], has become available for people with a history of dengue virus infection[12,13]. Since then, another live attenuated vaccine TAK-003 (QDENGA) has been authorized in several countries[14,15]. Finally, the Butantan–Dengue Vaccine (Butantan-DV) has attracted a great deal of interest following promising Phase 3 results[16].

Also, in recent years, a new type of DENV antiviral has been discovered using a phenotypic screen[17], specific to dengue and effective across all 4 serotypes. The first molecule of this series, JNJ A07, has been shown to display in vitro and in vivo efficacy and to be a highly potent NS4B-inhibitor. NS4B is a small, non-structural protein integrated into the membrane via 5 transmembrane domains. It associates with NS3 to form the scaffolding of the replication complex and is therefore indispensable to the viral cycle[18]. JNJ A07 acts by blocking the formation of the NS4B-based complex with NS3/NS2B that supports virus replication[19]. The molecule has nM pan-serotype and pan-genotype activity in vitro and is highly effective in vivo in mice. Two years after the characterization of JNJ A07, an analog optimized for clinical use, JNJ 1802 was produced[20]. This molecule, -renamed mosnodenvir for clinical trials- has a pM pan-serotype and genotype activity, and is also active in non-human primates against DENV-2 and DENV-1[20]. Mosnodenvir has completed a phase 1 clinical trial and is now in phase 2[21–24]. This phase 2 randomized multi-center clinical trial involves 37 different sites in South-East Asia, South America, Central America, and the Caribbean[22]. In the Caribbean, trials are taking place in Panama, Colombia, and Puerto Rico[22]. This last trial is the one closest to the French Caribbean Islands and started in 2023 with the first patient enrolled at the end of the month of February. No clinical trials have taken place or are underway in the French Caribbean Islands themselves. Both JNJ A07 and JNJ 1802 have a "high resistance barrier", with at least 20 passages (weeks) needed to obtain a fixed mutation conferring a significant decrease in sensitivity[19,20]. Selection pressure assays have shown that mutations at key positions within the NS4B induce escape by preventing the blockage of the NS3 interaction induced by both molecules. The main resistance-inducing mutations are T108I (termed "low resistance"), V91A, and L94F ("high resistance"). Mutations inducing the strongest reduction in sensitivity to compounds, in particular V91A, have previously been observed in the NS4B of wild-type strains of DENV-2 and DENV-3 at very low frequency[20].

The French National Reference Center (CNR) for arboviruses has implemented systematic genomic surveillance for all PCR-positive dengue cases referred to the center. To detect the emergence of DENV lineages potentially resistant to antiviral drugs in development, genomes are routinely screened for potential resistance mutations, including V91A and L94F. Here, by analyzing sequence data from dengue infections from the French Caribbean Islands[10], we show that the lineage responsible for the current epidemic exhibits the V91A mosnodenvir-resistance mutation. By evaluating the antiviral effect of JNJ A07 and JNJ 1802 (mosnodenvir) on multiple clinical isolates and on strains reconstituted using reverse genetics, we demonstrate that the epidemic lineage exhibits low sensitivity to these two molecules. Then, using a combination of phylogenetic analysis and in vitro resistance profiling, we show that V91A-carrying DENV lineages with likely low sensitivity to mosnodenvir emerged repeatedly over the last 20 years within serotype 2, but also in serotype 3.

## Results

### The lineage responsible for the current epidemic in the French Caribbean Islands exhibits low sensitivity to JNJ A07 and mosnodenvir

A previous phylogenetic analysis has shown that the epidemic in the French Caribbean Islands is associated with one main lineage belonging to the genotype II of DENV-2[10] (II.F.1). To determine whether some of the virus strains from the epidemic lineage could be resistant to mosnodenvir, we screened 98 DENV-2 genomes from cases recorded between February 2023 and January 2024 on the islands of Martinique, Guadeloupe, Saint Martin, and Saint Barthélemy (Supplementary Data 1) for previously described mutations conferring resistance to JNJ A07[19]. We found that the NS4B:V91A mutation (corresponding to a T to C transition at position 7001 in the CDS) was present in all DENV sequenced from the epidemic (Supplementary Table 1). This observation suggests that the lineage driving the epidemic in the French Caribbean Islands may be resistant to JNJ A07 and mosnodenvir.

Mutation V91A has been shown to induce a strong reduction in sensitivity to JNJ A07 and JNJ 1802 in vitro both (i) alone in a replicon system (genotype V - or Asian I), and (ii) combined with other resistance mutations including L94F and T108A in a competent virus (genotype I - or American)[19,20]. The decrease in sensitivity conferred by mutation V91A could thus be weaker if taken alone and in a genomic background distinct from the ones used to initially demonstrate its effect. Accordingly, we performed resistance profiling experiments for both JNJ A07 and JNJ 1802, using multiple virus isolates from different locations and at different times of the epidemic: April 2023 in Martinique (CNR_65877), December 2023 in Guadeloupe (CNR_72674), January 2024 in Saint-Barthelemy (CNR_73058) and March 2024 in Martinique (CNR_74234). We followed the same protocol as the one employed to characterize the pan genotype/serotype activity of the two molecules in vitro[19,20] and used as a reference CNR_25679, a DENV-2 strain from the Cosmopolitan genotype that lacks the V91A mutation, and for which sensitivity to JNJ A07 and JNJ 1802 (mosnodenvir) has already been described[19,20]. As a control, we added the adenosine analog NITD008[25]. This broad-spectrum flavivirus inhibitor targets the NS5 and should not be affected by NS4b mutations. Moreover, this compound has previously been characterized against DENV-2 CNR_25679 and others[26].

When measuring the activity of our control NITD008 against the four DENV-2 strains from the epidemic and CNR_25679, we obtained similar $EC_{50}$ values for all strains (mean fold change below 2), with values very close to what we had previously reported for this molecule[26], which validates our experimental protocol (Table 1). For JNJA07, we observed a very sharp decrease in antiviral activity with a mean fold change of 980 between the $EC_{50}$ values obtained with the four DENV-2-II strains carrying V91A (CNR_6587) and the one without V91A (CNR_25679) (Fig. 1, Table 1). Similarly, for JNJ 1802 (mosnodenvir), we observed a mean fold change of 33450 when comparing $EC_{50}$ values obtained with the four DENV-2-II strains and CNR_25679. For both molecules, these differences in $EC_{50}$ values translate into a drop in level of activity from pM to μM. Our results are in line with those already described for mutation V91A as well as those we previously reported for our reference strain DENV-2 CNR_25679[19,20] and show that across all tested virus isolates from the French Caribbean Islands epidemic (collected between April 2023 and March 2024), there is a drastic decrease in sensitivity to both JNJ A07 and JNJ 1802. Further, these findings suggests that all strains belonging to the French Caribbean islands epidemic lineage likely have low sensitivity to JNJ A07 and mosnodenvir as they all exhibit mutation V91A.

### Mutation V91A may have appeared shortly before the emergence of the lineage circulating in the French Caribbean Islands

Our sequence data analysis indicates that all strains from the French Caribbean Islands exhibit V91A, suggesting that the mutation may have

**Table 1 | Activity of JNJ A07, JNJ 1802, and NITD008 against DENV strains with and without the V91A mutation**

| Serotype | | 2 | | | | | | | 3 | |
|---|---|---|---|---|---|---|---|---|---|---|
| Genotype | | II | | | | | IV | | V | |
| Strain | | DENV-2 CNR_25679 | DENV-2 CNR_65877 V91A | DENV-2 CNR_72674 V91A | DENV-2 CNR_73058 V91A | DENV-2 CNR_74234 V91A | DENV-2/PG/BID-V2618/2008 | DENV-2/PG/BID-V2618/2008 V91A | DENV-3/BR/D3LIMHO/2006 | DENV-3/BR/D3LIMHO/2006 V91A |
| Origin/ Date | | PACA (Fr) 2014 | Martinique (Fr) 2023 | Guadeloupe (Fr) 2023 | Saint Barthelemy (Fr) 2024 | Martinique (Fr) 2024 | ISA strain | ISA strain | ISA strain | ISA strain |
| JNJ A07 | $EC_{50}$ (µM) | 0,000067 | 0,110 | 0,059 | 0,062 | 0,031 | 0,000069 | 0,200 | 0,00160 | 0,027 |
| | $EC_{90}$ (µM) | 0,000204 | 0,135 | 0,125 | 0,118 | 0,106 | 0,000252 | 0,824 | 0,004639 | 0,332 |
| | f.c | - | **1644,4** | **886,8** | **930,9** | **459,8** | - | **2892,9** | - | *23,0* |
| JNJ 1802 (mosnodenvir) | $EC_{50}$ (µM) | 0,000008 | 0,192 | 0,325 | 0,404 | 0,174 | 0,000081 | 0,081 | 0,00128 | 0,098 |
| | $EC_{90}$ (µM) | 0,000122 | 0,818 | 0,538 | 0,502 | 0,633 | 0,000176 | 0,198 | 0,008247 | 0,428 |
| | f.c | - | **23414,1** | **39783,2** | **49353,5** | **21261,9** | - | **1004,7** | - | *87,2* |
| NITD008 | $EC_{50}$ (µM) | 0,80 | 1,123 | 1,837 | 1,241 | 1,052 | 1,29 | 1,102 | 0,39 | 0,573 |
| | f.c | - | 1,4 | 2,3 | 1,6 | 1,3 | - | 0,9 | - | 1,5 |

Note: Fold changes above 100 are highlighted using bold fonts, and those comprised between 5 and 100 in italic fonts.
Interpolated $EC_{50}$ values are expressed in µM. $EC_{50}$ values are the mean for $n = 2$ to $n = 8$ independents experiments (detailed $EC_{50}$ and $EC_{90}$ values as source data in Supplementary Data 4). Fold change reductions were calculated in comparison with the $EC_{50}$ of each DENV genotype reference strain (with no V91A mutation). f.c: fold change.

emerged in a precursor of the epidemic lineage, the latter could have descendants circulating in other locations in the Caribbean or elsewhere. To evaluate if V91A is present in phylogenetic relatives of the French Caribbean Islands epidemic lineage, we combined the 98 DENV-2 sequences from the epidemic with all near-complete public genomes (>8500 nt) from the genotype II of DENV-2 and inferred their phylogenetic relationships (Figs. 2A and 2B). We found that mutation V91A was only present in the closest phylogenetic relative (0.002 substitution/site) of the French Caribbean Islands epidemic clade, a sequence sampled from a case in Florida in July 2023 (DENV-2/USA/FL-BPHL-0109/2023, Genbank accession: OR771147), which suggests that the mutation may have emerged recently. To try to identify additional phylogenetic relatives of the French Caribbean Islands epidemic clade, we downloaded all dengue sequences from the year 2023 and from January to March 2024, extracted sequences from genotype II and matched these sequences against our Genbank dataset to complement our DENV2 genotype II dataset. The phylogeny inferred based on this more comprehensive alignment did not allow us to identify other phylogenetic relatives than DENV-2/USA/FL-BPHL-0109/2023 (Supplementary Fig. 1). To estimate the time of emergence of V91A, we used a Bayesian approach to infer the time to the most recent common ancestor (TMRCA) of the French Caribbean Islands epidemic clade and DENV-2/USA/FL-BPHL-0109/2023 (Fig. 2C). We found that the TMRCA estimate was around April 1st 2022 (95% highest posterior density [HPD] interval: [2021-08-01:2022-11-26], Supplementary Table 2). This finding, together with the identification of the mutation in the closest phylogenetic relative of the French Caribbean Islands epidemic lineage, indicate that the emergence of V91A may have shortly preceded that of the epidemic lineage (estimated to be around September 30th 2022 (95% highest posterior density [HPD] interval: [2022-06-24:2022-12-26])[10].

### V91A-carrying DENV lineages with likely low sensitivity to mosnodenvir emerged repeatedly within serotype 2 over the last 30 years

Our phylogenetic analysis results indicate that the mosnodenvir resistance-inducing mutation V91A may have emerged in a precursor of the DENV-2 genotype II lineage that is associated with the current epidemic in the French Caribbean Islands. This is, however, not the first time that V91A is identified in circulating DENV lineages[20]. A previous study reported that 1.3% of the sequences available as of May 2020 on the Virus Pathogen Resource database (ViPR, now BV-BRC at https://www.bv-brc.org/), exhibited the mutation, which suggests that V91A-carrying lineages with likely low sensitivity to mosnodenvir other than the one associated with the French Caribbean Islands epidemic have circulated in the past –and may still circulate.

To determine whether –in addition to the current event– V91A has occurred once or multiple times within serotype 2 and to identify the genotypes compatible with the emergence of the mutation, we downloaded all DENV-2 sequences available on Genbank as of March 2024. When screening the sequences, we found that 37 exhibited the V91A mutation in addition to sequences from the French Caribbean Islands, its closest phylogenetic relative (OR821962), and another sequence from Florida (OR771147) likely corresponding to a potential import from the French Caribbean epidemic (Fig. 2). These sequences amount to ~0.86% of all 4658 publicly available sequences covering the NS4B region of DENV-2 (See Supplementary Data 2 and Supplementary Fig. 2). To determine the genotypes of these sequences, we combined V91A sequences with a set of reference sequences representative of DENV-2 genotypes and performed phylogenetic inference. We found that the 37 sequences carrying the V91A mutation were distributed across three genotypes: V (or Asian I: 30 sequences), II (or Cosmopolitan: 5 sequences), and III (or Asian American: 2 sequences) (Supplementary Fig. 2). These results show that V91A is compatible with at least three genotypes within the DENV-2 serotype.

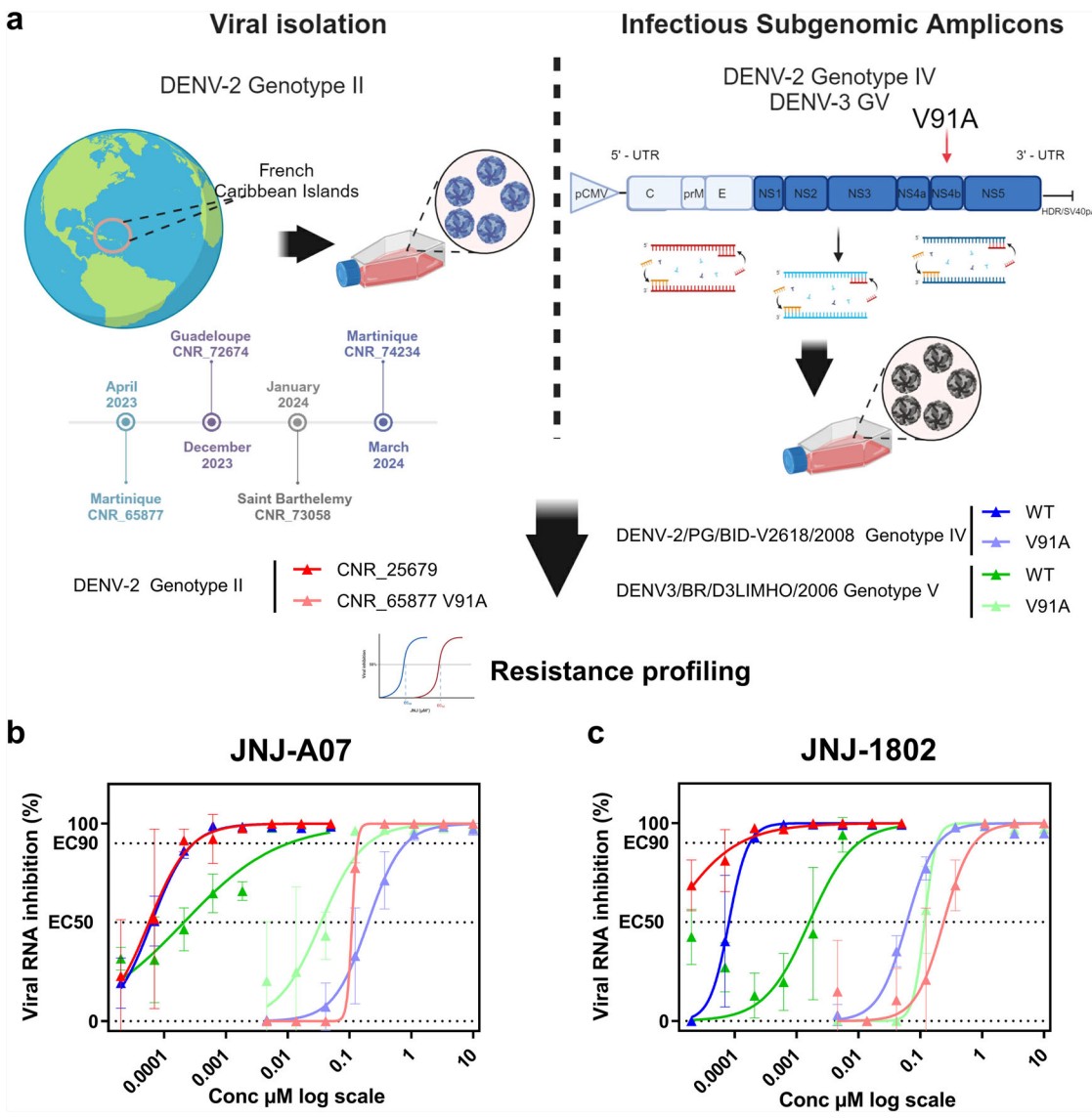

**Fig. 1 | In vitro characterization of V91A strains against JNJ A07 and JNJ 1802.**
**A** Strategy used to characterize the mutation V91A against JNJ A07 and 1802 NS4B inhibitors by testing either clinical samples or reverse-genetic strains obtained using the ISA strategy. Created in BioRender. Render3, B. (2024) BioRender.com/ u79r962. Dose response curves showing the susceptibility of DENV strains against **B** JNJ A07 and **C** JNJ 1802. Data presented are from a representative experiment performed in three technical replicates in VeroE6. Error bars show mean±s.d.

To evaluate the number of emergence events associated with V91A-carrying lineage(s), we built a maximum-likelihood phylogeny using all DENV-2 nearly complete genomes publicly available on Genbank as of March 29th 2024. We found that V91A has emerged at least on twelve occasions over the last 30 years within serotype 2 − in addition to the present event associated with the French Caribbean Islands epidemic (Fig. 3A). Most of the V91A sequences were isolated on the tree, suggesting the virus lineages exhibiting the mutation were either relatively short-lived, causing small magnitude transmission chains, or were largely undersampled. Six of the V91A sequences were recent, with sampling dates in the year 2019 or later. We could, however, identify three clades corresponding to larger transmission chains, that encompassed sequences sampled over several years. Two small clades encompassed two sequences each, one from Thailand, with sequences sampled in 2007 and 2010, and one from Viet-Nam with sequences sampled in 2022 and 2023. The third clade we identified comprised 21 sequences from Thailand, sampled in 2001, 2004, 2005, and 2014. These results show that V91A has emerged repeatedly over the last 30 years, with possible instances of circulation over

multiple years in addition to the present epidemic in the French Caribbean Island.

Previous studies[19,20] have shown that V91A results in a substantial decrease in sensitivity to both JNJ A07 and JNJ1802 in DENV-2 genomic backgrounds from genotype V (replicon system) and I (live virus) and our experiment identified a similar behavior with genotype II. These results suggest that V91A confers lower sensitivity to mosnodenvir across multiple –potentially all– genotypes of DENV-2[27]. To confirm the pan-DENV-2 effect of the V91A mutation, we evaluated its effect on sensitivity to JNJ A07 and JNJ 1802 (mosnodenvir) in a DENV-2 strain from genotype IV (DENV-2/PG/BID/V2618/2008). To produce a V91A-mutant of strain DENV-2/PG/BID/V2618/2008, we took advantage of the versatility of the Infectious Subgenomic Amplicons (ISA) reverse genetics method[28] which allows rapid rescue of single-stranded positive-sense RNA viruses in vitro by transfecting overlapping subgenomic DNA fragments. In this case, the DENV-2 system is composed of three fragments. We introduced V91A into the third subgenomic fragment of the system already developed for DENV-2/PG/BID/V2618/2008[26] –where NS4B is located. We then compared the profiles of

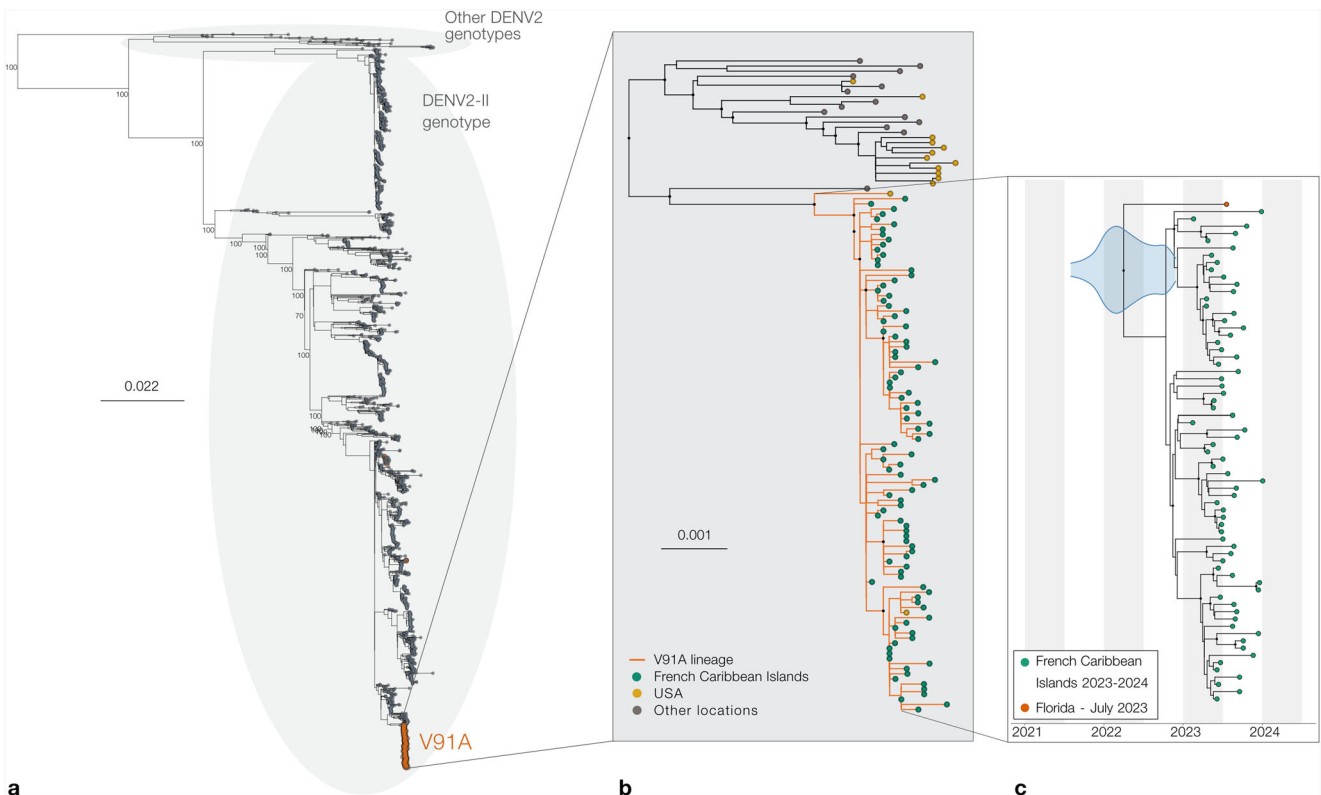

**Fig. 2 | V91A is present in the entire epidemic clade and its closest phylogenetic ancestor. A** Maximum-Likelihood phylogeny of all publicly available genomes from the French Caribbean Islands epidemic combined with all DENV2 sequences from the genotype II covering more than 85 % of the CDS available on Genbank as of March 2024 and a set of reference sequences from other genotypes. Phylogenetic inference was performed using IQTREE2 under a GTR + F + R5 substitution model with ultrafast bootstrap approximation (1000 replicates). The tree was rooted using the sylvatic genotype of DENV2. **B** In the subtree in the right-hand panel, sequences from the French Caribbean Islands epidemic are highlighted in sea green and sequences from the USA (Florida) are colored in mustard yellow, sequences from other locations are colored in gray. Branches are colored in orange for lineages exhibiting the V91A mutation. In the main tree, bootstrap values are shown for all nodes with more than 800 descendants. In the French Caribbean Islands subtree, all nodes with a boostrap support above 95 and more than 10 descendants are highlighted with a black circle. **C** Maximum clade credibility (MCC) tree of DENV genomes from the French Caribbean Islands epidemic and their closest phylogenetic ancestor, DENV-2/USA/FL-BPHL-0109/2023. Sequences from the French Caribbean Islands epidemic are highlighted in sea green and the sequence from the USA (Florida) is colored in mustard yellow. The blue violin plot indicates the 95% highest posterior density (HPD) interval for the tMRCA for the epidemic lineage and its closest phylogenetic relative. Nodes with posterior support above 0.9 are shown with a black dot.

sensitivity to JNJ A07 and JNJ 1802 of the genotype IV "wild type" strain and of its V91A mutant counterpart as we did previously for the two genotype II strains. For the DENV-2 genotype IV wild-type strain, we obtained a profile similar to that of the genotype II strain DENV-2 CNR_25679, with a high sensitivity of the wild-type to both molecules, with $EC_{50}$ values in the pM range (Fig. 1 and Table 1). Similarly, we observed a strong decrease in sensitivity with a fold change above 100 for the V91A mutant strain (Table 1), with $EC_{50}$ values in the μM range. Combined with our previous findings, these results show that V91A is able to confer low sensitivity to JNJ A07 and mosnodenvir across most genotypes of DENV-2.

**Mosnodenvir resistance-inducing mutation V91A also emerged previously in DENV-3, but has not been reported in DENV-1 or DENV-4**

In the previous section, we showed that V91A has been observed in other DENV-2 lineages than the French Caribbean Islands epidemic clade, with at least thirteen events of emergence events within DENV-2 serotype over the last 30 years. As V91A has also been reported in circulating lineages of DENV-3[20] we investigated DENV-1, DENV-3 and DENV-4 public sequence data to assess the emergence of V91A-lineages beyond serotype 2.

To determine the number of V91A emergence events within DENV serotype 3 and to identify the genotype(s) associated with the muta-tion, we downloaded all DENV-3 sequences available on Genbank as of

December 5th 2023 and screened them for the V91A mutation. In all, we identified 10 sequences carrying the V91A mutation, amounting to ~0.4% of all publicly available 2189 sequences covering the NS4B region of DENV-3 (Supplementary Table 3). Also, we identified one sequence carrying a V to L change at position 91 in the NS4B instead V to A. Using phylogenetic inference to determine the genotype of the DENV-3 sequences carrying mutation V91A, we found that they were equally distributed across genotypes II (5 sequences), and III (5 sequences) (Supplementary Fig. 3). The V91L sequence belonged to genotype II. To evaluate the number of emergence events associated with the DENV-3 V91A-carrying sequences, we built a phylogeny of all DENV-3 genomes publicly available on Genbank as of December 2023. We found that V91A emerged at least on 7 occasions over the last 30 years within serotype 3 (Fig. 3B). All but two of the V91A sequences were isolated on the tree, which suggests -as observed for DENV-2- that most lineages with the mutation either corresponded to small trans-mission chains or were undersampled. Only two sequences from Nicaragua from the years 2012 and 2013 grouped together in the phylogeny, which indicates a potential instance of lineage circulation over several months. Overall, these results show that V91A emerged multiple times within serotype 3 over the last 30 years and might have been associated with sustained virus transmission.

As we identified naturally occurring V91A-lineages within DENV-3 genotypes I and III, we sought to determine whether V91A also caused a decrease in sensitivity to JNJ A07 and JNJ 1802 (mosnodenvir) in a

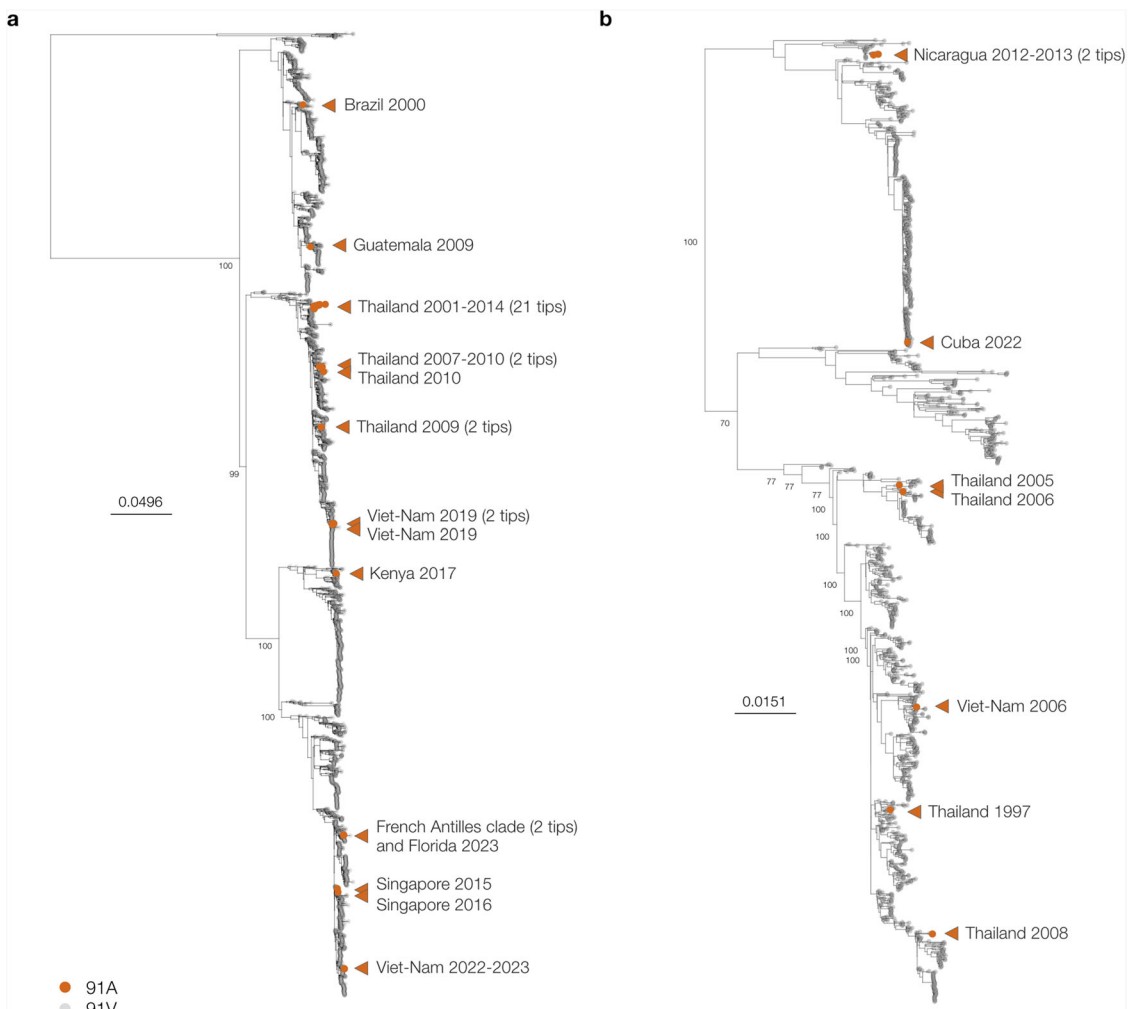

**Fig. 3 | V91A emergence events within DENV-2 (A) and DENV-3 (B) serotypes.** Maximum-Likelihood phylogeny of all publicly available genomes for DENV2 (a) and DENV-3 (b) covering more than 85 % of the CDS available on Genbank. Phylogenetic inference was performed using IQTREE2 under a GTR + F + R5 substitution model with ultrafast bootstrap approximation (1000 replicates). **A** The tree was rooted using the sylvatic genotype of DENV2. Sequences exhibiting the V91A mutation are highlighted in orange. Only one representative from the French

Caribbean Islands clade was included in the tree (in addition to the two sequences from the USA, one corresponding to the closest phylogenetic relative of the clade and the other to a potential import from the epidemic into Florida). Bootstrap values are shown for all nodes with more than 1280 descendants. **B** The tree was rooted using DENV1. Sequences exhibiting the V91A mutation are highlighted in orange. Bootstrap values are shown for all nodes with more than 500 descendants.

DENV-3 background. Using the same experimental setup as in the previous experiments, we evaluated the activity of both molecules against a DENV-3 strain from genotype V (DENV-3/BR/D3LIMHO/2006) and its V91A-mutant counterpart produced by reverse genetics. For the DENV-3 strain carrying mutation V91A, we observed $EC_{50}$ values similar (between 0.02 and 0.12) to those obtained for V91A-DENV-2 strains (all comprised between 0.08 and 0.8, Fig. 1 and Table 1). However for the wild-type DENV-3 strain, we found $EC_{50}$ values were higher (~0.001) than those obtained for wild-type DENV-2 strains (ranging between 0.00006 and 0.0004). These results indicate that in DENV-3, V91A does result in a reduction in sensitivity to both molecules, with fold changes between 18 and 60 (JNJ A07 and JNJ 1802, respectively), albeit with a weaker effect than that observed in DENV-2 − all fold changes were above 100 with DENV-2 strains (Table 1).

To determine if V91A circulating lineages have been identified within DENV serotypes 1 and 4, we downloaded all DENV-1 and DENV-4 sequences available on Genbank as of the 28th and 20th of November 2023, respectively, and screened them for the V91A mutation. Within DENV-1, none of the 5434 sequences exhibited an A at position 92 in the NS4B (equivalent of NS4B position 91 in DENV-2) but we identified 12 sequences with a V to I change at that position, all originating in

Thailand and collected between 2004 and 2006. Finally, we screened 968 DENV-4 sequences but none of them carried a V to A change at position 88 in the NS4B (equivalent of NS4B position 91 in DENV-2). These results suggest that V91A is not present in past or currently circulating lineages from DENV serotypes 1 and 4.

## Discussion

Due to its economic importance, public health burden and the number of people exposed, the development of countermeasures against dengue fever has been a priority over the last two decades. In addition to the significant progress made recently in the development of quadrivalent live attenuated vaccines, the identification of effective antivirals remains crucial, both for prevention in people who cannot be vaccinated or who respond poorly to the vaccine, and in people who are already infected, particularly if they are at risk of developing serious clinical forms, such as patients with sickle cell disease. In order to be effective and sustainable, antiviral therapies need to be implemented along with robust genomic surveillance of circulating strains i) to ensure that treatments are used against susceptible strains and ii) to monitor the emergence of resistance mutants and adjust treatment accordingly.

Recent experiments demonstrated that the activity of a promising dengue antiviral molecule, mosnodenvir, can be impaired by the presence of resistance mutations in NS4B in particular V91A and L94F[19,20]. Here, we show that the V91A mutation does indeed induce a strong reduction in sensitivity to this compound in DENV-2, using epidemic strains and a strain from genotype IV, engineered through reverse genetics[15]. Along with previous experiments[19,20], our results validated the ability of V91A to confer decreased sensitivity to mosnodenvir, alone, and across most genotypes of DENV-2 (II, III, IV, and V). The breadth of the effect of V91A on the sensitivity of strains from DENV-2 to mosnodenvir –low sensitivity was observed for four out of six genotypes– suggests the mutation might induce similarly low sensitivity to mosnodenvir in genotypes I and VI. However, these two genotypes have not been tested in this study as they are the least prominent in terms of epidemiology. The genotype I has been displaced by genotype III[24], and genotype VI is much less frequently reported than its counterparts[29–32].

As mutation V91A can be found in clinical isolates of DENV-2 but also DENV-3, we also evaluated the effect of V91A on sensitivity to mosnodenvir in this serotype. Interestingly, we only obtained a moderate decrease in sensitivity, with a fold change of 60 in $EC_{50}$ values. As V91A was obtained in DENV-2 resistance selection experiments, it is possible that the antiviral effect is in part specific to the topology of NS4B in this serotype and in DENV-3. Alternatively, this more moderate change in sensitivity could be explained by the naturally lower activity of mosnodenvir (and JNJ A07) against wild-type DENV-3[14], as the $EC_{50}$ values obtained with DENV-3-V91A and DENV-2-V91A strains were comparable. Interestingly, the $EC_{50}$ obtained for all the V91A strains ($0.2 < EC50 < 1\,\mu M$) are similar to the one obtained with closely related flaviviruses[20] WNV, JEV, and ZIKV highlighting the fact that the main antiviral mechanism based on blocking the NS4B/NS3 interaction is likely impaired, but that a less specific pan-flavivirus antiviral activity may remain.

When assessing the prevalence of V91A in DENV-3, we identified another type of amino acid change at position 91 in the NS4B, from a valine to a leucine. While this change in the residue is not equivalent to a valine-to-alanine shift (the valine side chain is longer than the alanine but shorter than the leucine), since it is located at the same position, it might also affect the sensitivity of DENV-3 strains to mosnodenvir and deserves further investigation. Similarly, while we did not identify the equivalents of V91A in any of the sequences publicly available for DENV-1 (V92A) and DENV-4 (V88A), we found 10 DENV-1 sequences carrying a valine to isoleucine change at the same position. As for DENV-3 V91L mutation, this mutation may not be equivalent to V91A but still deserves further investigation of its ability to confer reduced sensitivity to mosnodenvir.

In this study, we identify the mosnodenvir-resistance mutation V91A in an epidemic lineage of DENV-2 which suggests that the mutation is not detrimental to the replicative fitness or transmission cycle of the virus. We demonstrate that strains from this lineage exhibit a drastic reduction in sensitivity to the antiviral molecule with a more than 100-fold decrease in vitro. Although our findings have not yet been confirmed in vivo, they raise concerns regarding the efficacy of this antiviral molecule in patients infected with resistant strains. According to the pharmacokinetic results of the phase 1 study[21], to ensure the efficacy of the treatment, it would be necessary to maintain a concentration higher than the $EC_{90}$ of DENV-2 CNR_65877, i.e. $0.8\,\mu M$, which corresponds to $471\,ng/ml$ (similar EC values were observed for other DENV-2 from the French Caribbean Islands epidemic, V91A-DENV-2 genotype IV strain, and V91A-DENV-3 strain). The ongoing phase 2 clinical trial involves the use of two doses, a high dose consisting of a loading dose of 400 mg for two days, followed by a maintenance dose of 150 mg. The low dose consists of a loading dose of 150 mg for two days, followed by a maintenance dose of 50 mg[33]. Given mosnodenvir pharmacological profile, plasma concentrations

higher than 471 ng/ml will not be reached in the low-dose arm despite the observed plasma accumulation[21]. In the high-dose arm, the plasmatic concentration will reach and remain above 500 ng/ml for around ten days[21]. Once in vivo data is available for strains with low sensitivity to mosnodenvir, the results from the second arm of this trial may be useful to evaluate the clinical efficacy of mosnodenvir, originally developed with nM activity but on strains with µM activity.

While our work provides evidence that DENV-2 and DENV-3 strains with drastically reduced sensitivity to mosnodenvir have circulated and still circulated, for now, these strains likely constitute a minor fraction of circulating DENV strains. In 2023 in the Americas, the two only locations were the V91A lineage circulated in 2023 are the French Caribbean Islands and French Guiana – the epidemic lineage was shown to have expanded from the Caribbean to French Guiana[10] Among the more than 4.2 million cases reported in the Americas in 2023, only ~25,000 cases originated in the French Caribbean Islands[34] and ~2000 from French Guiana (where DENV-3 dominated for most of the year[34], which would represent less than 1% of reported cases in 2023 in the Americas. These estimates are obviously crude and complicated by heterogeneities in surveillance –both in terms of reported cases and of virus genomes sampled. On the public database BV-BRC (as of March 29th, 2024), there are currently less than 1000 sequences (>8500 nucleotides) published for the year 2023, our knowledge of the lineages that circulated that year is therefore still extremely partial. Nevertheless, based on the data at hand –and disregarding the potential presence of other resistance mutations– most circulating dengue strains should be sensitive to mosnodenvir, which thus remains a crucial tool in the fight against dengue.

The findings from this study re-emphasize the need for robust genomic surveillance as we show that DENV lineages with decreased sensitivity to mosnodenvir likely circulated in the recent past and still circulate. Specifically, thorough monitoring is urgently needed during clinical trials for antiviral molecules, to allow for a clear assessment of their effect in relation to the expected sensitivity of circulating strains based on their genomic sequence. The ongoing circulation of a lineage of DENV-2 with low sensitivity to mosnodenvir in an epidemic context in the French Caribbean Islands is of particular concern. First, the Lesser Antilles have been identified as a source of DENV dissemination, seeding other American countries in the past[35]. Second, there is phylogenetic evidence that the lineage in question both became established on the American continent (French Guiana), and initiated autochthonous circulation events in Europe (France)[10]. Third, the closest phylogenetic relative of the V91A French Caribbean Islands epidemic lineage is a sequence from Florida, which suggest that other related lineages carrying V91A may circulate elsewhere in the Caribbean –the large volume of travel between Florida and the Caribbean islands make it a good sentinel for the region[36–38]. Finally, in this study we also identified V91A-sequences from 2023 in Asia (Viet-Nam Fig. 2), which suggest V91A lineage(s) might currently circulate in that region. However, based on the only sequence available, it is challenging to assess this last point in reliable terms at present.

With the exceptional dengue activity reported in recent years, and with 2024 shaping up to be even more devastating than previous years[39], ensuring the effectiveness of the tools used to control this disease appears more crucial than ever. Here, we provided evidence of large-scale transmission of a lineage with low sensitivity to mosnodenvir in the French Caribbean Islands, which could severely limit the effectiveness of this antiviral treatment in similar epidemics in the future. By combining phylogenetic inference, reverse-genetics and in vitro experiments, we show that lineages with likely reduced sensitivity to mosnodenvir emerged repeatedly in the recent past, indicating a clear risk of circulation of resistant DENV-2 and DENV-3 strains in the future. Our findings warrant for increased genomic tracking of circulating dengue lineages globally, and provide an example of how genomic surveillance and in vitro resistance profiling can be effectively

combined to investigate the circulation of virus strains with low sensitivity to antivirals.

## Methods

### Cell line
VeroE6 cells (CLC 1586) were obtained from ATCC and were grown in MEM (Minimal Essential Medium-Life Technologies) with 7.5% heat-inactivated Fetal Calf Serum (FCS; Life Technologies with 1% penicillin/streptomycin PS, 5000U.mL$^{-1}$ and 5000 µg.mL$^{-1}$ respectively (Life Technologies) and supplemented with 1% non-essential amino acids (Life Technologies), at 37 °C with 5% $CO_2$.

### Viral strains description
Genotype II strain DENV-2 CNR_25679 (France, GenBank: MF004385, EVAg Ref-SKU:001V-02229), genotype IV strain BID-V2618 (Papua New Guinea, GenBank: FJ906959.1, EVAg Ref-SKU:001V-03106), and genotype V strain DENV3/BR/D3LIMHO/2006 (Brazil, GenBank: JN697379.1, EVAg Ref-SKU: 001V-03108) were phylogenetically described elsewhere[26] and were previously used to characterize JNJ A07[11] and JNJ 1802[15].

Strains used in this study from the DENV-2 epidemic in the French Caribbean: DENV-2 CNR_65877 (formerly described as Mart-2023) whose sample was taken in Martinique (France) in April 2023, DENV-2, DENV-2 CNR_72674 whose sample was taken in Guadeloupe (France) in December 2023, DENV-2 CNR_73058 whose sample was taken in St Barthelemy (France) in January 2024 and DENV-2 CNR_74234 whose sample was taken in Martinique (France) in March2024. All strains were isolated by inoculating 100 uL of sample residual from diagnostics on a confluent culture of Vero E6 on 6-well flat bottom cell culture plates. The inoculum was incubated 1 hour at 37 °C in a 5% CO2 atmosphere to infect cells monolayers prior to be removed and replaced by 4 mL of MEM supplemented with 7% heat-inactivated FBS, 1% penicillin–streptomycin, 1% L-glutamine, 1% Kanamycin, and 3% Amphotericin B. Cell cultures were examined daily for the potential presence of cytopathic effect (CPE). At post-inoculation day 5, supernatants were aliquoted and used for RNA extraction and virus detection by RT-qPCR.

### Virus propagation
To prepare our working stocks, a 25 cm2 culture flask of confluent Vero E6 cells growing with MEM medium with 2.5 % FBS (Life Technologies) was inoculated with 100 µl of infectious supernatant. Cell supernatant medium was harvested at the peak of infection and supplemented with 25 mM HEPES (Sigma) before being stored freeze in small aliquots at −80 °C.

All experiments with replicative viruses were performed in a BSL3 laboratory.

All virus strains are available for the community at European Virus Archive. https://www.european-virus-archive.com/.

### Antiviral compounds
JNJ A07 was purchased from MedChemexpress, JNJ-1802 was purchased form Probechem and NITD008 was purchased from Hit2lead (www.hit2lead.com).

### RNA extraction and quantification
Viral RNA was performed as previously described[19,20,26] using a QIAamp viral RNA kit on the automated QIAcube (Qiagen) following manufacturer recommendations. Relative quantification of viral RNA was performed as previously described[19,20,26] using the GoTaq® 1-Step RT-qPCR System kit (Promega). The mixture contained 5 µL of 2x Master Mix, 0.25 µL of each primer (250 nM), 0.07 µL of probe (75 nM), 0.2 µL of GoScript RT Mix and 3.8 µL of extracted nucleic acids. Assays were performed using the QuantStudio 12 K Flex real-time PCR machine (Life technologies). Synthetic RNA was used to calculate the amount of viral RNA from standard curves. Set of primers and probes used for RT-qPCR have been already described[26].

### RNA quantification in clinical samples
RT-qPCR tests were performed on Panther Fusion (HOLOGIC) using a dual-target assay that targets a conserved region in 3' UTR[40]. The Ct values obtained for all clinical samples are detailed in Supplementary Table 4.

### EC$_{50}$ and EC$_{90}$ determination
One day prior to infection, 5×104 VeroE6 cells per well were seeded in 100 µL assay medium (containing 2.5% FBS) in 96 well culture plates. The next day, antiviral compounds (JNJ A07, JNJ 1802 and NITD008 were added using the D300e dispenser (TECAN) with eight ½ dilutions. JNJ A07 and JNJ 1802 two types of experiments were performed one with "low dose" to determine wild type strains EC50 and EC90 and one with high dose to determine strains carrying V91A EC50 and EC90. Then, 25 µL/well of a virus mix diluted in the medium was added to the wells. Prior to the assay it was verified for each DENV strain that they were harvested during the logarithmic growth phase of viral replication at 96 hours post infection[19,20,26]. Four virus control wells were included within the plate. Quantification of DENV genome by real-time RT-qPCR was done as described above. Viral inhibition was calculated as follow: 100* (quantity mean VC- sample quantity)/ quantity mean VC. The 50% effective concentrations (EC50 compound concentration required to inhibit viral RNA replication by 50%) and the 90% effective concentrations (EC90 compound concentration required to inhibit viral RNA replication by 90%) were determined using logarithmic interpolation after performing a nonlinear regression (log(inhibitor) vs. response–Variable slope (four parameters)). All data obtained were analyzed using GraphPad Prism 9 software (Graphpad software).

### Virus recovery by infectious subgenomic amplicons (ISA)
DNA preparation was performed from de novo synthetized gene (Genscript) using the same PCR conditions, primers and purification method as previously reported[26]. The ISA procedure for virus recovery was performed as previously described[26,27]. Briefly, one-day prior to transfection, a mix of BHK21 and HEK293 cells was seeded on a 96 wells amine pure coat plate (Corning) in order to reach sub-confluence on the day of transfection. Then, 200 ng of an equimolar mix of the three DNA fragments (sub genomic amplicons from the DNA preparation) were transfected using lipofectamine 3000 (Life Technologies), following the manufacturer's instruction in 15 replicates. 6 days later, the supernatant was passed on a standard 96 well plate (Greiner) containing VeroE6 cells. Then, 4 days after infection viral supernatant was collected and analyzed by qRT-PCR to assess viral replication.

### Virus sequence datasets
All publicly available sequences for dengue virus were downloaded for each of DENV serotypes from the NCBI Nucleotide database, Genbank (keywords: "Dengue virus 4" database accessed in November 2023; "Dengue virus 1" database accessed on November, 2023; "Dengue virus 3" database accessed in December, 2023; "Dengue virus 2" database accessed on March, 2023). We filtered the data by: (i) excluding sequences that did not belong to DENV species, (ii) excluding sequences corresponding to patents, recombinants, clones. The remaining sequences were aligned using MAFFT (version 7.511[41]), trimmed to their coding regions (ORF) and inspected manually. We kept only the sequences encompassing the region of the NS4B protein encompassing the V91A mutation (for serotypes 1 and 2), or its equivalent for serotypes 3 (92) and 4 (88). Final datasets included 5434 sequences of DENV1, 4658 sequences of DENV2, 2189 datasets of DENV3, and 968 sequences of DENV4. We also generated a set comprising exclusively DENV2-II sequences (length > 8500 nt, 1828), and a set corresponding to the 98 sequences from the French Caribbean

Islands current epidemic available on Genbank as of March 2024. Finally, we also included DENV-2 sequences from the GISAID database that were not present on Genbank to ensure the comprehensiveness of our results, the corresponding accession numbers are listed in Supplementary Data 3.

## Prevalence analysis

We used the four datasets comprising all DENV-1, DENV-2, DENV-3 and DENV-4 sequences to screen for the the V91A mutation or its equivalent for DENV1 (V92A) and DENV4 (V88A). For DENV-2 and DENV-3 sequences carrying the V91A mutation, we determined their genotype by combining them with reference sequences representative of DENV-2 and DENV-3 genotypes and inferring a maximum-likelihood phylogeny with IQ-Tree (version 1.6.12, [8-9]), using the best-fit model identified by ModelFinder and assessed branch support using an ultrafast bootstrap approximation (UFBoot2) (1000 replicates). The sets of reference genomes used for genotyping were selected based on the set of genomes used in the widely-used tool genome detective to DENV genotyping[42]. Additionally, for the set of reference genomes to better reflect the phylogenetic diversity within DENV2 serotype, we added the sequence of the highly divergent strain QML22 isolated from a traveler returning from Borneo to Australia[43]. For the set of reference genomes to better reflect the phylogenetic diversity within DENV3 serotype, we added three sequences representative of genotype IV (L11433, L11434, L11439), as well as three sequences representative of genotype V (KM190937, JN697379, AF317645).

## Inference of phylogenies for DENV-2, DENV-2-II, and DENV-3

We first filtered out all sequences with a CDS length below 8500 nucleotides from the DENV-2 and DENV-3 datasets and then removed potential recombinant sequences from the DENV-2, DENV-2 Comopolitan and DENV-3 datasets using the Recombination Detection Program (RDP) version 4. We used RDP, GENCONV, and MAXCHI methods for primary screening and BOOTSCAN and SISCAN methods to check for recombination signals[44-48]. We used the automask option to ensure optimal recombination detection. V91A-carrying DENV-2 sequence OQ028215 and DENV-3 sequences KF921927 and MZ008477 were identified as potential recombinant and thus removed from downstream analyses.

Using the resulting recombinant-free alignments, we performed ML phylogenetic reconstruction with IQ-Tree (version 1.6.12[49]), using a GTR + F + R5 model (General time reversible model with empirical base frequencies and a FreeRate model with 5 categories) and assessed branch support using an ultrafast bootstrap approximation (UFBoot2) (1000 replicates).

## Addition of GISAID data

DENV-2 sequences from GISAID were genotyped using phylogenetic inference as described above and those corresponding to the DENV2-II genotype were added to the DENV2-II dataset before performing recombination detection and ML phylogenetic reconstruction as described above.

## Bayesian inference of a time-resolved phylogeny

To evaluate the timing of emergence of the V91A mutation in the precursor of the lineage circulating in the French Caribbean Islands, we reconstructed time-scaled phylogenies with BEAST (v1.10.5[50]). We first performed a root-to-tip analysis and removed sequences that were too partial or whose sampling date was incongruent with their genetic divergence resulting in a subset of 69 sequences from the French Caribbean Islands 2023 exhibiting sufficient association between genetic distances and sampling dates ($R^2 = 0.58$, Supplementary Fig. 4) to perform bayesian inference. We used two different substitution models (the HKY substitution model with a gamma-distributed rate variation among sites and no partition into codon

positions (HKYG4), or the Shapiro-Rambaut-Drummond-2006 (SRD06)), a uncorrelated lognormal (UCLN) clock model clock, and three different coalescent models (constant, exponential and bayesian skygrid). For the constant and exponential coalescent models, we ran single MCMC chains of 50 million states with the BEAGLE computational library[51]. For the bayesian skygrid coalescent model, we ran single MCMC chains of 100 million states. We used Tracer (v1.7[52]) for inspecting the convergence and mixing, discarding the first 10 % of steps as burn-in, and ensuring that estimated sampling size (ESS) values associated with estimated parameters were all >200. To identify the best fitted model we performed marginal likelihood estimation using path sampling/ stepping-stone sampling. All alignment, xml, and tree files for this study are available at: https://github.com/Snseli/Dengue_resistance_supplementary.git.

## Reporting summary

Further information on research design is available in the Nature Portfolio Reporting Summary linked to this article.

## Data availability

All alignment, xml and tree files for this study are available at: https://github.com/Snseli/Dengue_resistance_supplementary.git (https://doi.org/10.5281/zenodo.13680149)[53]. All unique sequence IDs are detailed in the Supplementary Information and on the github repository listed above. Source data are provided with this paper as a source data file in Supplementary Data 4. Virus isolates used in this study can be made available on the European Virus Archive Global portal (www.european-virus-archive.com) upon request.

## Code availability

All script and xml files for this study are available on github (https://github.com/Snseli/Dengue_resistance_supplementary.git, https://doi.org/10.5281/zenodo.13680149)[53].

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

## Acknowledgements

We thank Magali Gilles (UVE; Marseille), Emilie Kaufmann (UVE; Marseille) and Camille Placidi (UVE; Marseille) for their valuable technical contribution. We thank all collaborators from the associated National Reference Centers in French Guiana (Dominique Rousset, Alisé Lagrave, Anne Lavergne, Antoine Enfissi) and in Réunion (Marie-Christine Jaffar-Bandjee, Etienne Frumence, Nicolas Traversier), from hospital and research centers in the French Caribbean Islands (André Cabié, George Dos Santos, Laurence Fagour, Raymond Césaire) and in Réunion (Patrick Gérardin), from Santé publique France (Lucie Fournier, Lucie Leon, Frédérique Dorléans, Muriel Vincent), and from the arbovirus genomics diagnostic laboratories working group for they contribution to DENV genomic surveillance efforts. A part of the work was done on the Aix Marseille University antivirals platform "MaSC". This work was supported by the ARBOGEN project, funded by the MSDAVENIR Foundation. The activity of the National Reference Centers is supported by the French national health agency (Santé publique France). The activity of the UVE is supported by its supervising institutional bodies (Aix-Marseille Université, Università di Corsica, Institut national de la santé et de la recherche médicale, Institut de recherche pour le développement, Institut de Recherche Biomédicale des Armées). HSW PhD fellowship is supported by IRD. This work was supported by the IRD Chair "Antiviral strategy for emergence in the South", in partnership with Aix-Marseille University, Inserm and ANRS I MIE.

## Author contributions

Conceptualization: F.T., R.K., and X.dL. Methodology: H.S.B., S.S., G.P., R.A., T.C., M.G., A.F., N.A., and L.P. Formal analysis: F.T., R.K., S.S., R.A., and G.P. Investigation: F.T. and R.K. Resources: F.T., R.K., N.A., L.P., G.G., G.A.D., and A.F. Writing original draft: F.T., R.K. X.dL. Writing review & editing: all co-authors. Visualization: F.T. and R.K. Funding acquisition: F.T., R.K., X.dL., G.G., and G.A.D.

## Competing interests

The authors declare no competing interests.
