## [Peer Review File · Nature Communications]

Genomic surveillance reveals a dengue 2 virus epidemic lineage with a marked decrease in sensitivity to MosnodenvirREVIEWER COMMENTS

Reviewer #1 (Remarks to the Author):

Review of "Genomic surveillance reveals that the dengue 2 virus lineage responsible for the 2023-2024 epidemic in the French Caribbean Islands is resistant to Mosnodenvir"

This is an interesting manuscript describing the common occurrence of drug resistant variants in circulating virus against an inhibitor that is still in clinical trials.

Genomic variants associated with resistance were observed and reverse genetics experiments were performed to confirm that viruses with isolated mutations as well as mutations in circulating virus backgrounds show resistance.

Good use of a non-related RDRP inhibitor as a control.

Good use of phylogenetics to estimate the recent occurrence of TMRCA within just a few years (2021) in the cosmopolitan dengue clade.

What is not clear is that the significance of commonly occurring resistance against a drug that is still in clinical trials warrants publication in Nature Communications. Perhaps especially true since this is not the first description of naturally occurring viruses with resistance causing mutations.

I need to point out here that a clear and universal naming system for dengue clades is sorely needed. What is cosmopolitan now will be replaced by something new at some point, likely soon. And naming viruses or clades of virus with reference to where they originate or circulate is unethical.

Minor suggestions:

Consider removing the first "is" in the following sentence:

"Interestingly, the EC50 obtained for all the V91A strains ($0.2 < EC50 < 1\mu M$) are similar to the one obtained with closely related flaviviruses 16 WNV, JEV and ZIKV highlighting the fact that the main antiviral mechanism is based on blocking the NS4B/NS3 interaction is likely impaired, but that a less specific pan-flavivirus antiviral activity may remain."

More substantial suggestions:

Are any dengue vaccines in use in the study area?

Is there a mosnodenvir clinical trial occurring in the study area?

Figure 1 shows virus being isolated from a patient blood draw. Is there an associated IRB approval that should be included?

Testing one clinical isolate and then claiming all strains from epidemic are likely resistant since they carry the mutation sequence. Should screen multiple isolates to better uphold this claim? Graphs show virus resistance to drugs, clearly requiring increased concentration of drugs with mutated virus, but mutated viruses still reach 100% inhibition. Do the highest concentrations surpass levels used in clinical trials? Perhaps need to re-word the "resistance" to tone down the effect/meaning since the drug may still work.

Adding in virus for EC50/90 assays without known titers? But authors say the viruses were harvested during log phase. Is it necessary to know titer input for each virus to make sure the amount is similar among all viruses used for resistance evaluation? Or not necessarily needed because resistance was judged by RNA decrease via qPCR?

In the intro and results the authors have a skeptical perspective on the drug mosnodenvir, with an emphasis of importance on surveillance and potential problems using mosnodenvir for treatment because there is some resistance to the drug, but not complete. Then toward the discussion, the claims seem to be reversed with support for the drug as it can still work. Suggest a more consistent emphasis throughout the manuscript.

Suggest showing a time resolved tree with the TMRCA results.

Suggest including two citations when discussing recent sequences from Florida:

Jones FK, Morrison AM, Santiago GA, Rysava K, Zimler RA, Heberlein LA, et al. Introduction and Spread of Dengue Virus 3, Florida, USA, May 2022–April 2023. *Emerg Infect Dis.* 2024;30(2):376-379. <https://doi.org/10.3201/eid3002.231615>

Taylor-Salmon, E., Hill, V., Paul, L.M. et al. Travel surveillance uncovers dengue virus dynamics and introductions in the Caribbean. *Nat Commun* 15, 3508 (2024). <https://doi.org/10.1038/s41467-024-47774-8>

Reviewer #2 (Remarks to the Author):

In their analysis of dengue virus (DENV) genomes from the current epidemic in the French Caribbean Islands, Bouzidi et al. reveal that all examined genomes exhibit the NS4B:V91A mutation, which is associated with significant resistance to mosnodenvir in vitro.

The article is well-crafted and timely, aligning with the global context of mosquito-borne diseases. Establishing a surveillance strategy in high-risk regions is crucial for developing tailored preventive measures. The authors' extensive efforts in experimental virology and the analysis of the effects of specific point mutations are commendable. These findings underscore the need for enhanced genomic surveillance, especially to monitor strains with resistance mutations, which could improve the efficacy assessment of DENV treatments currently under development.

Suggestions for improvement:

The title seems overly lengthy and could be more impactful. I suggest that the authors consider revising it to be more concise and engaging.

The Methods section mentions RNA quantification, but lacks details about the methodology used and the quantity of RNA in the sampled samples. Additional information would be beneficial.

More information is needed on Infectious Subgenomic Amplicons. Including such details would enhance the clarity and depth of the study.

The criteria for selecting reference genomes and the rationale for discarding certain samples should be clarified.

Given the study's focus on mutational patterns, more comprehensive details on the detection of specific point mutations are necessary.

Regarding the Bayesian inference for constructing a time-resolved phylogeny, it is unclear if the authors tested for temporal signal and whether the dataset was suitable for this analysis. Further elaboration on these points would be helpful.

Reviewer #3 (Remarks to the Author):

Reviewer #4 (Remarks to the Author):

Bouzidi and collaborators investigated a 2023-24 Dengue epidemic in the French Caribbean islands using genomic surveillance. During this study they found a single major DENV2 lineage bearing a specific NS4B mutation (V91A) known to significantly reduce the effectiveness of Mosnodenvir, one Dengue antiviral undergoing phase II clinical trials. However, this mutation was first characterized in other lineages/genotypes genomic context, so the authors used an array of assays to evaluate the antiviral resistance impact of this mutations in currently circulating lineages of DENV2 and 3. While the mutation induced a significant antiviral efficacy reduction in the DENV2 backbone it almost did not impact DENV3 replication in cell culture. Lastly, the authors accessed publicly available Dengue sequences from all four serotypes to evaluate how frequently this mutation emerged in the past and its current prevalence. They found that this mutation emerged independently multiple times, but that genomes bearing such mutations did not successfully spread widely, which is different from the findings in the French Caribbean islands. These findings are timely and as far as I'm aware, report for the first time the natural emergence of mutations that impact the efficacy of the Mosnodenvir efficacy. Although this antiviral is still not available for widespread use, knowing that resistance mutations emerged in the past and are frequently found now in some outbreaks such as the French Caribbean island outbreak is crucial to plan proper

pharmacological intervention in clinical practice in this region. Moreover, it is imperative to keep increasing the molecular surveillance efforts in neighboring countries and around the globe to closely monitor the spread of such DENV2 lineage or the emergence of other lineages that may render the antiviral treatment with Mosnodenvir ineffective in other regions of the globe. Follow below some points that should be addressed by the authors.

Comments:

Introduction - correct "Ae. Albopictus" to "Ae. albopictus".

Page 2 line 46-47 - "The islands of Guadeloupe, Martinique, Saint Martin and Saint Barthélemy all entered a state of epidemic during the second half of 2023, with more than 30,000 cases as of March 2024." What was the incidence per 1000 inhabitants? It would be interesting to bring here the human population size of these islands as well.

Page 2 line 69 - please clarify what it means "high resistance barrier"? That the molecules are resilient to resistance mutations on the viral genome? If yes, what experiments and data support this claim?

Page 2 - It would be important to add how many doses of the Dengue vaccines had been administered in the French Caribbean islands if any and where the Mondesivir clinical trials are being conducted - in the Caribbean islands? Related to that, in the discussion section, it would be interesting to highlight the need of molecular surveillance (targeted or genomic) on the populations that will be treated with this compound in the future.

Table 1 - I found DENV-2 CNR_25679 in the table and CNR-25679 in the text. Standardize using underline or dash.

Page 3 lines 117-122 - Regarding the fold change numbers presented in the text and Table 1. I'm not seeing a correspondence of these values. For instance for JNJA07 it was reported a higher than 1900 fold change for EC50 in the text comparing cosmopolitan Mart-2023 and CNR-25679, but in the table I'm seeing a value of 1664.4 of fold change. Maybe the fold change calculation presented in the table is an average of EC50 and EC90? Please clarify.

Page 4 line 152 - 153 - add the HPD 95% density curves as a section of Figure 2. Besides, giving the HPD 95% of the clade including FL-BPHL and the estimates for the French Caribbean Islands last common ancestor I think the data does not support a clear early emergence of the lineage before the French Caribbean Islands outbreak. I suggest the authors review this section and conclusion.

Page 8 line 195 - The authors stated "and of small magnitude or largely undersampled" change to "and caused small magnitude transmission chains or were largely undersampled"

Page 8 line 197 - The authors stated "three clades of larger magnitude" change to "three clades that underwent larger transmission chains".

Page 10 line 233 - The authors stated "with at least thirteen events of emergence 233 events in DENV-2 over the last 30 years" change to "with at least thirteen independent emergence within the DENV-2 serotype over the last 30 years"

Page 10 line 243 - The authors stated "we combined 243 them with a set of reference sequences representative of DENV-3 genotypes and performed 244 again a phylogenetic inference." This is a bit repetitive once it was done for DENV3 and DENV2 serotypes. Describe it only once in the MM section.

For supplementary figure 1 and 2 it is confusing to have color for genotypes and for the presence of a given mutation. Keep colors for genotypes and add different symbols for sequences bearing or not the mutation.

Figure 3 and 4 can be combined in a multi section figure.

Page 13 line 297 - The authors stated "ii) that their use does not lead to the emergence of resistance mutants." I suggest changing to "ii) to monitor the emergence of resistance mutants and adjust treatment" once resistant strains are likely to emerge when there is enough selection pressure (antiviral treatment).

Page 13 line 306 - The authors stated "was observed four out of six genotypes" correct to "was observed for four out of six genotypes"

Page 13 line 312 - "As mutation V91A mutation" correct.

The author can also use the most up to date Dengue data from EpiArbo within GISAID to have access to a more comprehensive dataset of DENV genomes.

Page 18 line 501-502 - "a uncorrelated 501 lognormal (UCLN) clock model clock, and a bayesian skygrid coalescent model." Did the authors perform molecular clock model and coalescent models evaluation (stepping stone sampling) before selecting these for analysis? If not, it is a must. Moreover, it is also important to access the molecular clockness of the dataset for instance using TempEst.

Gabriel da Luz Wallau

Answer:

Reviewer #1 (Remarks to the Author):

Review of “Genomic surveillance reveals that the dengue 2 virus lineage responsible for the 2023-2024 epidemic in the French Caribbean Islands is resistant to Mosnodenvir”

This is an interesting manuscript describing the common occurrence of drug resistant variants in circulating virus against an inhibitor that is still in clinical trials.

Genomic variants associated with resistance were observed and reverse genetics experiments were performed to confirm that viruses with isolated mutations as well as mutations in circulating virus backgrounds show resistance.

Good use of a non-related RDRP inhibitor as a control.

Good use of phylogenetics to estimate the recent occurrence of TMRCA within just a few years (2021) in the cosmopolitan dengue clade.

What is not clear is that the significance of commonly occurring resistance against a drug that is still in clinical trials warrants publication in Nature Communications. Perhaps especially true since this is not the first description of naturally occurring viruses with resistance causing mutations.

Answer: We thank the reviewer for the positive assessment of our work. Regarding the reviewer's concern relative to the significance of our findings, we would like to point out that (1) among antiviral drugs against dengue virus (DENV) currently in development, mosnodenvir is the most promising as it has progressed the furthest in clinical trials (PMID: 34698303) and is currently in phase 2 in multiple locations (Janssen, <https://globaltrialfinder.janssen.com/trial/CR109157>). Also, (2), while clinical isolates carrying mutation V91A have been identified in the past, this is the first time the mutation is found in a lineage responsible for an epidemic. Finally, (3) we assess for the first time, the sensitivity to mosnodenvir of clinical isolates carrying the V91A mutation, showing they become much less sensitive to the drug. Our results raise crucial issues for DENV control strategies, highlighting the potential limitation of the use of mosnodenvir and the need for genomic surveillance to screen for potentially antiviral resistant DENV lineages including those carrying the V91A mutation.

I need to point out here that a clear and universal naming system for dengue clades is sorely needed. What is cosmopolitan now will be replaced by something new at some point, likely soon. And naming viruses or clades of virus with reference to where they originate or circulate is unethical.

Answer: We fully agree with the reviewer on the urgent need for establishing a clear, standardized nomenclature system for dengue lineages. Hence, in the revised version of the manuscript, we adapted our strain naming system so that (1) genotypes and lineages follow the recommendation of Hill and colleagues (PMID: 38798319), which we think provides a strong basis for the systematic naming of DENV lineages, and (2) replaced the Mart-2023 strain name with the anonymized number corresponding to the sample analyzed at the French National Reference Centre for Arboviruses (CNR) and from which the virus isolate was obtained.

Minor suggestions:

Consider removing the first "is" in the following sentence:

"Interestingly, the EC₅₀ obtained for all the V91A strains ($0.2 < EC_{50} < 1 \mu M$) are similar to the one obtained with closely related flaviviruses 16 WNV, JEV and ZIKV highlighting the fact that the main antiviral mechanism is based on blocking the NS4B/NS3 interaction is likely impaired, but that a less specific pan-flavivirus antiviral activity may remain."

Answer: We corrected the sentence accordingly.

More substantial suggestions:

Are any dengue vaccines in use in the study area?

Answer: We thank the reviewer for this question. Concerning the use of dengue vaccine in the area, there is none use in the French Caribbean Islands, but one clinical trial in pregnant women has started in Puerto Rico in 2022 with the DENG VAXIA from Sanofi, with a follow-up planned until 2026 (<https://classic.clinicaltrials.gov/ct2/show/NCT04486638?term=vaccine&cond=Dengue&draw=3&rank=126#contacts>).

Is there a mosnodenvir clinical trial occurring in the study area?

Answer: We thank the reviewer for highlighting this question that is key to providing the full context of our study. We checked the 37 sites where a mosnodenvir phase 2 clinical trial is taking place (Janssen, <https://globaltrialfinder.janssen.com/trial/CR109157>), and none of them is located in the French Caribbean Islands - the closest one began in Puerto Rico in February 2023. We added this information in the introduction and mention the issue of resistance in the context of clinical trials in the discussion (see below). While we cannot fully rule out a scenario in which the French Caribbean islands epidemic V91A-carrying lineage would have emerged in Puerto Rico at the beginning of the mosnodenvir clinical trial but the timeline of the epidemic makes this scenario unlikely. First, the French Caribbean epidemic appears to have started prior to the launching of the Mosnodenvir clinical trial in Puerto Rico, with the increase in cases first noted in November 2022. As all the strains from the epidemic identified to date correspond to the V91A-carrying lineage, this would suggest that the lineage in question emerged before the start of the Puerto Rico clinical trial. This timeline is further confirmed by our molecular clock analysis that dates the emergence of the V91A lineage back to 2022-04-01 (95% HPD: 2021-08-01-14;2022-11-26, updated in response to one of the reviewer's comments).

Lines 68-76

"This phase II randomized multi-center clinical trial involves 37 different sites in South-East Asia, South America, Central America and the Caribbean. In the Caribbean, trials are taking place in Panama, Colombia and Puerto Rico. This last trial is the one closest to the French Caribbean Islands and started in 2023 with the first patient enrolled at the end of the month of February. No clinical trials have taken place or are underway in the French Caribbean Islands themselves."

Lines 373-378

"The findings from this study re-emphasize the need for robust genomic surveillance as we show that DENV lineages with decreased sensitivity to mosnodenvir likely circulated in the recent past and still circulate. Specifically, thorough monitoring is urgently needed during clinical trials for antiviral molecules, to allow for a clear assessment of their effect in relation to the expected sensitivity of circulating strains based on their genomic sequence."

Figure 1 shows virus being isolated from a patient blood draw. Is there an associated IRB approval that should be included?

Answer: This study was part of the national public health surveillance program of the National Reference Centre (NRC) for Arboviruses supervised by the National Public Health Agency (Santé Publique France, SPF). Therefore, as an epidemiological record, consultation with an ethics committee was not required. Samples transferred to the NRC are submitted with a form that includes the patient's consent (non-opposition clause). Human samples and cultures originating from human samples were anonymised, with no or minimal risk to patients within the terms of the European General Data Protection Regulation and the French National Commission on Informatics and Liberty (CNIL).

Testing one clinical isolate and then claiming all strains from epidemic are likely resistant since they carry the mutation sequence. Should screen multiple isolates to better uphold this claim?

Answer: We agree with the reviewer on this insightful observation, and in response we have added three other isolates from different locations and sampling times, as follows: one from the island of Guadeloupe/December 2023, one from the island of Saint-Barthélemy/January 2024 and finally another from the island of Martinique/March 2024, i.e. 11 months after the one we originally studied. We performed three tests (n=3) on each isolate: two under 'high concentration' conditions and one under 'low concentration' conditions, as described in the Materials and Methods section. All these results confirm our initial observations and are reported in Table 1, and details of the individual EC50/EC90s are included in the source file that is associated with the manuscript.

Graphs show virus resistance to drugs, clearly requiring increased concentration of drugs with mutated virus, but mutated viruses still reach 100% inhibition. Do the highest concentrations surpass levels used in clinical trials? Perhaps need to re-word the “resistance” to tone down the effect/meaning since the drug may still work.

Answer: We thank the reviewer for this excellent remark. We agree that the term resistance may suggest that any inhibition from the molecule is eliminated while virus replication can still be inhibited to some extent. Whether the concentrations required to prevent virus replication in real-life conditions remains, however, to be determined and we address this point in the discussion (see below). Also, to ensure that we do not use terms that may mislead the reader, we reformulated the title and the text, replacing resistance to more cautious formulations e.g., “decreased sensitivity”

Lines 344-357

“According to the pharmacokinetic results of the phase 1 study¹⁷, to ensure the efficacy of the treatment, it would be necessary to maintain a concentration higher than the EC90 of DENV-2

CNR_65877, i.e. 0.8 μM , which corresponds to 471 ng/ml (similar EC values were observed for other DENV-2 from the French Caribbean Islands epidemic, V91A-DENV-2 genotype IV strain, and V91A-DENV-3 strain). The ongoing phase 2 clinical trial involves the use of two doses, a high dose consisting of a loading dose of 400 mg for two days, followed by a maintenance dose of 150 mg. The low dose consists of a loading dose of 150 mg for two days, followed by a maintenance dose of 50 mg²⁹. Given mosnodenvir pharmacological profile, plasma concentrations higher than 471 ng/ml will not be reached in the low-dose arm despite the observed plasma accumulation. In the high-dose arm, the plasmatic concentration will reach and remain above 500 ng/ml for around ten days.”

Adding in virus for EC50/90 assays without known titers? But authors say the viruses were harvested during log phase. Is it necessary to know titer input for each virus to make sure the amount is similar among all viruses used for resistance evaluation? Or not necessarily needed because resistance was judged by RNA decrease via qPCR?

Answer: This is a good observation and indeed the titration in FFU or TCID50 is not mandatory for our viral RNA yield reduction assay. Our assay did not depend on the cytopathogenic potential of the strain, thus allowing for the rapid inclusion of any dengue strain. Because all these strains could differ in their replication kinetics, prior to the assay, all DENV inputs and times of readout of the assay were calibrated so that the replication levels were still in the logarithmic part of the growth curve at time of the collection of the supernatant. Although the maximum reduction of virus yield may depend of the specific strain and assay conditions, the half effective concentration (EC50s) are not expected to be affected in these settings and will depend only on inhibitor efficiency. Each assay is verified and validated using the NITD008 control compound which has a pan flavivirus activity and should not be affected by NS4B mutations. Moreover, this RNA yield reduction assay was previously used for evaluating the pan-genotype activity of JNJ A07 (Kaptein et al., 2021) and JNJ 1802 (Goethals et al., 2023) , facilitating the direct comparison of our results with those from previous studies.

In the intro and results the authors have a skeptical perspective on the drug mosnodenvir, with an emphasis of importance on surveillance and potential problems using mosnodenvir for treatment because there is some resistance to the drug, but not complete. Then toward the discussion, the claims seem to be reversed with support for the drug as it can still work. Suggest a more consistent emphasis throughout the manuscript.

Answer: We are sorry that the reviewer finds our view on the drug mosnodenvir “skeptical”. Without further details it is hard for us to nail down the elements that suggest so in the text. The simple structure of the text may artificially convey a shift from a “skeptical” view to a more favorable one as we start with a thorough assessment of the risk of emergence of V91A-associated resistance to mosnodenvir and then put our results back into context - emphasizing that the risk we identified

remains confined to a limited number of cases. But we aim for our work to be objective and for our views. A more balanced version of the results could have been obtained by including a formal evaluation of the number of DENV cases that could be attributable to V91A DENV lineages in 2023, but as we were limited by the data available, we had to move that evaluation to the discussion section. Following the reviewer's advice, we performed one proofreading session dedicated to making the text more consistent and clear.

In addition, in response to another reviewer's comment we also reformulated our results and discussion to refer to our strain as exhibiting decreased sensitivity to mosnodenvir rather than being resistant to reflect the fact that the drug still inhibits the replication of V91A strains, albeit at much higher concentrations that may not be compatible when using this treatment in humans.

Suggest showing a time resolved tree with the TMRCA results.

Answer: We thank the reviewer for this suggestion. We've now added a plot of the maximum-clade-credibility tree obtained by bayesian inference using BEAST that also shows the posterior distribution obtained for the age of the root estimates in Figure 2.

Suggest including two citations when discussing recent sequences from Florida:

Jones FK, Morrison AM, Santiago GA, Rysava K, Zimler RA, Heberlein LA, et al. Introduction and Spread of Dengue Virus 3, Florida, USA, May 2022–April 2023. *Emerg Infect Dis.* 2024;30(2):376-379. <https://doi.org/10.3201/eid3002.231615>

Taylor-Salmon, E., Hill, V., Paul, L.M. et al. Travel surveillance uncovers dengue virus dynamics and introductions in the Caribbean. *Nat Commun* 15, 3508 (2024). <https://doi.org/10.1038/s41467-024-47774-8>

Answer: We thank the reviewer for this suggestion. We've now added these two references in the discussion (see below).

Line 383-386

"Third, the closest phylogenetic relative of the V91A French Caribbean Islands epidemic lineage is a sequence from Florida, which suggest that other related lineages carrying V91A may circulate elsewhere in the Caribbean —the large volume of travel between Florida and the Caribbean islands make it a good sentinel for the region."

Reviewer #2 (Remarks to the Author):

In their analysis of dengue virus (DENV) genomes from the current epidemic in the French Caribbean Islands, Bouzidi et al. reveal that all examined genomes exhibit the NS4B:V91A mutation, which is associated with significant resistance to mosnodenvir in vitro.

The article is well-crafted and timely, aligning with the global context of mosquito-borne diseases. Establishing a surveillance strategy in high-risk regions is crucial for developing tailored preventive measures. The authors' extensive efforts in experimental virology and the analysis of the effects of specific point mutations are commendable. These findings underscore the need for enhanced genomic surveillance, especially to monitor strains with resistance mutations, which could improve the efficacy assessment of DENV treatments currently under development.

Answer: We thank the reviewer for this positive assessment of our work.

Suggestions for improvement:

The title seems overly lengthy and could be more impactful. I suggest that the authors consider revising it to be more concise and engaging.

Answer: We agree that the title is a little bit long. We've now changed it to a more concise formulation.

"Genomic surveillance reveals a dengue 2 virus epidemic lineage with a marked decrease in sensitivity to Mosnodenvir"

The Methods section mentions RNA quantification, but lacks details about the methodology used and the quantity of RNA in the sampled samples. Additional information would be beneficial.

Answer: We thank the reviewer for pointing out this need for additional details in the methods section. We now detail the RTqPCR system and protocol used for molecular detection in the methods section and provide Ct values for the samples in supplementary table 5.

"RT-qPCR tests were performed on Panther Fusion (HOLOGIC) using a dual-target assay that targets a conserved region in 3' UTR. The Ct values obtained for all clinical samples are detailed in Supp Table 5."

More information is needed on Infectious Subgenomic Amplicons. Including such details would enhance the clarity and depth of the study.

Answer: We agree with the reviewer that our original manuscript was lacking information about the Infectious Subgenomic Amplicons strategy so we added details relative to the method in the results section :

Line 226-231

“To produce a V91A-mutant of strain DENV-2/PG/BID/V2618/2008, we took advantage of the versatility of the Infectious Subgenomic Amplicons (ISA) reverse genetics method²³ which allows rapid rescue of single-stranded positive-sense RNA viruses *in vitro* by transfecting overlapping subgenomic DNA fragments. In this case, the DENV-2 system is composed of three fragments. We introduced V91A into the third subgenomic fragment of the system already developed for DENV-2/PG/BID-V2618/2008²²—where NS4B is located.”

We also expanded the ISA method description in the material and methods section:

Line 470-479

“DNA preparation was performed from de novo synthesized gene (Genscript) using the same PCR conditions, primers and purification method as previously reported²². The ISA procedure for virus recovery was performed as previously described^{22,33}. Briefly, one-day prior to transfection, a mix of BHK21 and HEK293 cells was seeded on a 96 wells amine pure coat plate (Corning) in order to reach sub-confluence on the day of transfection. Then, 200 ng of an equimolar mix of the three DNA fragments (sub genomic amplicons from the DNA preparation) were transfected using lipofectamine 3000 (Life Technologies), following the manufacturer’s instruction in 15 replicates. 6 days later, the supernatant was passed on a standard 96 well plate (Greiner) containing VeroE6 cells. Then, 4 days after infection viral supernatant was collected and analyzed by qRT-PCR to assess viral replication.”

The criteria for selecting reference genomes and the rationale for discarding certain samples should be clarified.

Answer: We thank the reviewer for highlighting the need for clarification of the sequence selection strategy. The sets of reference genomes used for sequence genotyping were selected based on a paper written by Fonseca and colleagues which describes the reference genomes used by the widely-used tool genome detective to DENV genotyping (Fonseca et al. 2019). Additionally, for the set of reference genomes to better reflect the phylogenetic diversity within DENV2 serotype, we added the sequence of the highly divergent strain QML22 isolated from a traveler returning from Borneo to Australia (Liu et al. 2016). For the set of reference genomes to better reflect the phylogenetic diversity within DENV3 serotype, we added three sequences representative of genotype IV (L11433, L11434, L11439), as well as three sequences representative of genotype V (KM190937, JN697379, AF317645). Finally, for bayesian inference, following this comment and another reviewer’s comment, we performed a new round of selection of sequences for bayesian inference and now describe the root-to-tip analysis as

part of the methods section and provide the root-to-tip plot as a supplementary figure. We expect this selection to increase the robustness of the TMRCA estimate.

Lines 504-511

“The sets of reference genomes used for genotyping were selected based on the set of genomes used in the widely-used tool genome detective to DENV genotyping (Fonseca et al. 2019). Additionally, for the set of reference genomes to better reflect the phylogenetic diversity within DENV2 serotype, we added the sequence of the highly divergent strain QML22 isolated from a traveler returning from Borneo to Australia (Liu et al. 2016). For the set of reference genomes to better reflect the phylogenetic diversity within DENV3 serotype, we added three sequences representative of genotype IV (L11433, L11434, L11439), as well as three sequences representative of genotype V (KM190937, JN697379, AF317645).”

Lines 528-532

“We first performed a root-to-tip analysis and removed sequences that were too partial or whose sampling date was incongruent with their genetic divergence resulting in a subset of 69 sequences from the French Caribbean Islands 2023 exhibiting sufficient association between genetic distances and sampling dates ($R^2=0.58$, Supp. Figure 4) to perform bayesian inference.”

Given the study's focus on mutational patterns, more comprehensive details on the detection of specific point mutations are necessary.

Answer: We thank the reviewer for pointing out this need for additional methodological details. To screen DENV sequences for the presence of mutations associated with reduced sensitivity to Mosnodenvir, we used an in-house python script that is now provided on github (https://github.com/Snseli/Dengue_resistance_supplementary.git) as a supplementary material. Briefly, this script translates all sequences from a multiple sequence alignment (containing exclusively coding sequences), and screens them for the presence of any of the following mutations in the NS4B: F47Y, S85L, V91A, L94F, P104S, T108I, A137T, T216N, T216P.

Regarding the Bayesian inference for constructing a time-resolved phylogeny, it is unclear if the authors tested for temporal signal and whether the dataset was suitable for this analysis. Further elaboration on these points would be helpful.

Answer: We agree with the reviewer that our methods section lacked details regarding the tests performed to confirm the suitability of our dataset for the inference of a time-resolved phylogeny. Following this comment and another reviewer's comment, we performed a new round of selection of sequences for bayesian inference and now describe the root-to-tip analysis as part of the methods section and provide a root-to-tip plot as a supplementary figure.

Lines 528-532

“We first performed a root-to-tip analysis and removed sequences that were too partial or whose sampling date was incongruent with their genetic divergence resulting in a subset of 69 sequences from the French Caribbean Islands 2023 exhibiting sufficient association between genetic distances and sampling dates ($R^2=0.58$, Supp. figure 4) to perform bayesian inference.”

Reviewer #3 (Remarks to the Author):

Answer: We thank the reviewer for participating in the reviewing of our work which has allowed us to improve the quality of the manuscript substantially.

Reviewer #4 (Remarks to the Author):

Bouzidi and collaborators investigated a 2023-24 Dengue epidemic in the French Caribbean islands using genomic surveillance. During this study they found a single major DENV2 lineage bearing a specific NS4B mutation (V91A) known to significantly reduce the effectiveness of Mosnodenvir, one Dengue antiviral undergoing phase II clinical trials. However, this mutation was first characterized in other lineages/genotypes genomic context, so the authors used an array of assays to evaluate the antiviral resistance impact of this mutations in currently circulating lineages of DENV2 and 3. While the mutation induced a significant antiviral efficacy reduction in the DENV2 backbone it almost did not impact DENV3 replication in cell culture. Lastly, the authors accessed publicly available Dengue sequences from all four serotypes to evaluate how frequently this mutation emerged in the past and its current prevalence. They found that this mutation emerged independently multiple times, but that genomes bearing such mutations did not successfully spread widely, which is different from the findings in the French Caribbean islands. These findings are timely and as far as I'm aware, report for the first time the natural emergence of mutations that impact the efficacy of the Mosnodenvir efficacy. Although this antiviral is still not available for widespread use, knowing that resistance mutations emerged in the past and are frequently found now in some outbreaks such as the French Caribbean island outbreak is crucial to plan proper pharmacological intervention in clinical practice in this region. Moreover, it is imperative to keep increasing the molecular surveillance efforts in neighboring countries and around the globe to closely monitor the spread of such DENV2 lineage or the emergence

of other lineages that may render the antiviral treatment with Mosnodenvir ineffective in other regions of the globe. Follow below some points that should be addressed by the authors.

Answer: We thank the reviewer for this positive assessment of our work.

Comments:

Introduction - correct "Ae. Albopictus" to "Ae. albopictus".

Answer: This term has been corrected according to the reviewer's suggestion.

Page 2 line 46-47 - "The islands of Guadeloupe, Martinique, Saint Martin and Saint Barthélemy all entered a state of epidemic during the second half of 2023, with more than 30,000 cases as of March 2024." What was the incidence per 1000 inhabitants? It would be interesting to bring here the human population size of these islands as well.

Answer: We agree that adding the incidence per 1000 inhabitants would provide a clearer picture of the scale of the epidemic. Based on the last report from the National Institute of Statistics and Economic Studies (INSEE), the population is approximately 360,749 in Martinique(<https://www.insee.fr/fr/statistiques/7739431>), 384,315 in Guadeloupe(<https://www.insee.fr/fr/statistiques/7739221>), and 31,801 and 10,457 in Saint Martin and in Saint Barthélemy, respectively(<https://www.insee.fr/fr/statistiques/7704076>), leading to a total population of approximately 787,322. As of March 2024, the incidence would thus be above 38 cases per 1000 inhabitants, which is high, in comparison, the incidence reported for last year in Brazil was 1359 cases per 100 000 population that, is ~13 cases per 1 000 inhabitants. We've now included a sentence summarizing this point in the introduction.

Lines 45-48

"A case count that amounts to a high incidence — above 38 cases per 1000 inhabitants given the islands latest population counts⁷⁻⁹. In comparison, the incidence reported for last year in Brazil was 1359 cases per 100 000 population¹ that, is ~13 cases per 1 000 inhabitants."

Page 2 line 69 - please clarify what it means "high resistance barrier"? That the molecules are resilient to resistance mutations on the viral genome? If yes, what experiments and data support this claim?

Answer: The term comes from the first two descriptions of the molecules (Kaptein et al., 2021 and Goethals et al., 2023) where resistance barrier experiments were performed, and at least 20

passages (week) were needed to obtain a fixed, strong, resistance mutation (Kaptein et al., 2021, Extended Data Fig. 3: Time-of-drug-addition and in vitro resistance selection). So this term means that when exploring the molecule resistance barrier, it is high because you cannot quickly obtain a resistance mutation due the selection pressure applied by the compound. We've now reformulated the text to make this term less ambiguous.

Line 73-75

“Both JNJ A07 and JNJ 1802 have a “high resistance barrier”, with at least 20 passages (week) needed to obtain a fixed mutation conferring a significant decrease in sensitivity”

Page 2 - It would be important to add how many doses of the Dengue vaccines had been administered in the French Caribbean islands if any and where the Mondesivir clinical trials are being conducted - in the Caribbean islands? Related to that, in the discussion section, it would be interesting to highlight the need of molecular surveillance (targeted or genomic) on the populations that will be treated with this compound in the future.

Answer: We thank the reviewer for this question, concerning the use of dengue vaccines in the area, we found that none of them had been used to date in the French Caribbean Islands. One clinical trial for DENGVAXIA (Sanofi) is however, taking place in Puerto Rico (<https://classic.clinicaltrials.gov/ct2/show/NCT04486638?term=vaccine&cond=Dengue&draw=3&rank=126#contacts>). Regarding the phase 2 clinical trial for Mosnodenvir, none of the 37 sites listed by the promoter (Janssen, <https://globaltrialfinder.janssen.com/trial/CR109157>), is located in the French Caribbean Islands, the closest one is taking place in Puerto Rico and began in February 2023. We added information regarding ongoing trials in the introduction and now discuss the question of surveillance during clinical trials in greater detail in the discussion.

Lines 68-73

“This phase II randomized multi-center clinical trial involves 37 different sites in South-East Asia, South America, Central America and the Caribbean. In the Caribbean, trials are taking place in Panama, Colombia and Puerto Rico. This last trial is the one closest to the French Caribbean Islands and started in 2023 with the first patient enrolled at the end of the month of February. No clinical trials have taken place or are underway in the French Caribbean Islands themselves.”

Lines 373-378

“The findings from this study re-emphasize the need for robust genomic surveillance as we show that DENV lineages with decreased sensitivity to mosnodenvir likely circulated in the recent past and still circulate. Specifically, thorough monitoring is urgently needed during clinical trials for antiviral

molecules, to allow for a clear assessment of their effect in relation to the expected sensitivity of circulating strains based on their genomic sequence.”

Table 1 - I found DENV-2 CNR_25679 in the table and CNR-25679 in the text. Standardize using underline or dash.

We thank the reviewer for pointing out this discrepancy, it has been corrected.

Page 3 lines 117-122 - Regarding the fold change numbers presented in the text and Table 1. I'm not seeing a correspondence of these values. For instance for JNJA07 it was reported a higher than 1900 fold change for EC50 in the text comparing cosmopolitan Mart-2023 and CNR-25679, but in the table I'm seeing a value of 1664.4 of fold change. Maybe the fold change calculation presented in the table is an average of EC50 and EC90? Please clarify.

We thank the reviewers for this careful reading of our manuscript, it was a typo, as we have added new strains to the manuscript, we now are reporting the mean fold change of all V91A strains from the Caribbean epidemic compared to the wild type.

Page 4 line 152 - 153 - add the HPD 95% density curves as a section of Figure 2. Besides, giving the HPD 95% of the clade including FL-BPHL and the estimates for the French Caribbean Islands last common ancestor I think the data does not support a clear early emergence of the lineage before the French Caribbean Islands outbreak. I suggest the authors review this section and conclusion.

Following the suggestion of the reviewer, we added the HPD density curve to the time resolved-tree added in Figure 2 in response to the reviewer's comment. We agree with the reviewer that there is an alternative hypothesis to the emergence of V91A prior to the emergence of the lineage associated with the epidemic in the French Caribbean Islands, which would imply i) the emergence of the mutation following the beginning of the epidemic with no sampling of the initial lineage and the divergence of the lineage corresponding to the FL-BPHL sequence after this event of emergence. While this hypothesis is less likely because it would require an absence of sampling of the non-V91A lineage during the epidemic it cannot be excluded with the data available as there is an overlap between the 95%HPD intervals of the time of emergence of the V91A mutation and that of the French Caribbean Islands epidemic lineage. We thus reformulated our text accordingly (see below).

Section header

“Mutation V91A may have appeared shortly before the emergence of the lineage circulating in the French Caribbean Islands”

Results

Lines 167-171

“This finding, together with the identification of the mutation in the closest phylogenetic relative of the French Caribbean Islands epidemic lineage, indicate that the emergence of V91A may have shortly preceded that of the epidemic lineage (estimated to be around September 30th 2022 (95% highest posterior density [HPD] interval: [2022-06-24:2022-12-26])⁶.”

Line 184-186

“Our phylogenetic analysis results indicate that the mosnodenvir resistance-inducing mutation V91A may have emerged in a precursor of the DENV-2 genotype II lineage that is associated with the current epidemic in the French Caribbean Islands.”

Page 8 line 195 - The authors stated “and of small magnitude or largely undersampled” change to “and caused small magnitude transmission chains or were largely undersampled”

The sentence has been changed according to the reviewer’s suggestion.

Page 8 line 197 - The authors stated “three clades of larger magnitude” change to “three clades that underwent larger transmission chains”.

Following the reviewer’s advice, the text has been updated to a new formulation (see below).

“three clades corresponding to larger transmission chains”

Page 10 line 233 - The authors stated “with at least thirteen events of emergence 233 events in DENV-2 over the last 30 years” change to “with at least thirteen independent emergence within the DENV-2 serotype over the last 30 years”

The text has been changed according to the reviewer’s suggestion.

Page 10 line 243 - The authors stated “we combined 243 them with a set of reference sequences representative of DENV-3 genotypes and performed 244 again a phylogenetic inference.” This is a bit repetitive once it was done for DENV3 and DENV2 serotypes. Describe it only once in the MM section.

Following the reviewer’s advice, we removed the repetition in the results section. The new formulation can be found below.

Line 262-264

“Using phylogenetic inference to determine the genotype of the DENV-3 sequences carrying mutation V91A, we found that they were equally distributed across genotypes II (5 sequences), and III (5 sequences) (Supp. Figure 3).”

For supplementary figure 1 and 2 it is confusing to have color for genotypes and for the presence of a given mutation. Keep colors for genotypes and add different symbols for sequences bearing or not the mutation.

We thank the reviewer for pointing out this issue with the coloring scheme of the supplementary figures 2 and 3 (former supplementary figures 1 and 2). They have been updated so that, at every tip of the tree, the center of the circle is colored according to the genotype and the outline of the circle is colored according to the presence/absence of the mutation. This new version should convey the results of our genotyping analysis more clearly.

Figure 3 and 4 can be combined in a multi section figure.

The figures have been combined following the reviewer’s advice.

Page 13 line 297 - The authors stated “ii) that their use does not lead to the emergence of resistance mutants.” I suggest changing to “ii) to monitor the emergence of resistance mutants and adjust treatment” once resistant strains are likely to emerge when there is enough selection pressure (antiviral treatment).

Following the reviewer’s advice, the sentence has been revised.

“ii) monitor the emergence of resistance mutants and adjust treatment accordingly.”

Page 13 line 306 - The authors stated “was observed four out of six genotypes” correct to “was observed for four out of six genotypes”

The sentence has been changed accordingly.

Page 13 line 312 - "As mutation V91A mutation" correct.

The sentence has been corrected.

The author can also use the most up to date Dengue data from EpiArbo within GISAID to have access to a more comprehensive dataset of DENV genomes.

Following the reviewer's advice we attempted to complement our DENV2-II dataset with DENV2 sequence data from the years 2023-2024 available on GISAID, as additional data may provide further information on the origin and dispersal of the French Caribbean Islands epidemic lineage. As sequences pulled from Genbank are not identifiable on GISAID we had to compare all GISAID DENV2 entries from the years 2023 and 2024 to those in our current dataset, considering to have a match if the sequence, sampling date and sampling location were identical, due to different conventions on both platforms, this procedure did not allow to remove all sequences from Genbank present in the GISAID dataset so we built a first tree to remove all those remaining duplicates. After removal, we built a final tree including all Genbank and GISAID sequences up to March 2024 and found that our conclusions were left unchanged, with 1 additional sequence present in the epidemic clade, likely corresponding to an import to Brazil. We now mention this additional analysis in the results section of the manuscript.

Lines 157-163

"To try to identify additional phylogenetic relatives of the French Caribbean Islands epidemic clade, we downloaded all dengue sequences from the year 2023 and from January to March 2024, extracted sequences from genotype II and matched these sequences against our Genbank dataset to complement our DENV2 genotype II dataset. The phylogeny inferred based on this more comprehensive alignment did not allow us to identify other phylogenetic relatives than DENV-2/USA/FL-BPHL-0109/2023 (Supp Figure 1)."

Page 18 line 501-502 - "a uncorrelated 501 lognormal (UCLN) clock model clock, and a bayesian skygrid coalescent model." Did the authors perform molecular clock model and coalescent models evaluation (stepping stone sampling) before selecting these for analysis? If not, it is a must. Moreover, it is also important to assess the molecular clockness of the dataset for instance using TempEst.

Thank you for pointing out this need for additional details. We performed our initial analysis based on a dataset previously used for bayesian inference and for which we had performed temporal signal

assessment and best model selection (Klitting et al. 2024). Following the reviewer's comment, we performed a new round of selection from our French Caribbean Islands dataset, assessed the presence of a temporal signal in the data using TempEst and found that the association between the genetic distances and their sampling dates ($R^2=0.58$, Supp Figure 4) to infer a time-resolved phylogeny in BEAST. For Bayesian inference, we used the best model identified in a prior evaluation of the best coalescent model using both stepping-stone and path sampling (Klitting et al. 2024). Following the reviewer's comment we performed another round of evaluation of the best coalescent model using both stepping-stone and path sampling on the present data which confirmed that the Shapiro-Rambaut- Drummond-2006 (SRD06) substitution model with a lognormal (UCLN) clock and a bayesian skygrid coalescent model was best to use for inference as detailed below. Our manuscript has been updated with the results of this new analysis and that details of the root-to-tip analysis and best model evaluation.

Lines 528-543

"We first performed a root-to-tip analysis and removed sequences that were too partial or whose sampling date was incongruent with their genetic divergence resulting in a subset of 69 sequences from the French Caribbean Islands 2023 exhibiting sufficient association between genetic distances and sampling dates ($R^2=0.58$, Supp Figure 4) to perform bayesian inference. We used two different substitution models (the HKY substitution model with a gamma-distributed rate variation among sites and no partition into codon positions (HKYG4), or the Shapiro-Rambaut-Drummond-2006 (SRD06)), a uncorrelated lognormal (UCLN) clock model clock, and three different coalescent models (constant, exponential and bayesian skygrid). For the constant and exponential coalescent models, we ran single MCMC chains of 50 million states with the BEAGLE computational library⁴². For the bayesian skygrid coalescent model, we ran single MCMC chains of 100 million states. We used Tracer (v1.743) for inspecting the convergence and mixing, discarding the first 10 % of steps as burn-in, and ensuring that estimated sampling size (ESS) values associated with estimated parameters were all >200. To identify the best fitted model we performed marginal likelihood estimation using path sampling/ stepping-stone sampling. All alignment, xml, and tree files for this study are available at: https://github.com/Snseli/Dengue_resistance_supplementary.git."

Gabriel da Luz Wallau

References

Fonseca, Vagner, Pieter J. K. Libin, Kristof Theys, Nuno R. Faria, Marcio R. T. Nunes, Maria I. Restovic, Murilo Freire, et al. 2019. "A Computational Method for the Identification of Dengue, Zika and Chikungunya Virus Species and Genotypes." *PLoS Neglected Tropical Diseases* 13 (5): e0007231.

Hill, Verity, and Guy Baele. 2019. "Bayesian Estimation of Past Population Dynamics in BEAST 1.10 Using the Skygrid Coalescent Model." *Molecular Biology and Evolution* 36 (11): 2620–28.

Klitting, Raphaëlle, Géraldine Piorkowski, Dominique Rousset, André Cabié, Etienne Frumence, Alisé Lagrave, Anne Lavergne, et al. 2024. "Molecular Epidemiology Identifies the Expansion of the DENV2 Epidemic Lineage from the French Caribbean Islands to French Guiana and Mainland France, 2023 to 2024." *Euro Surveillance: Bulletin Europeen Sur Les Maladies Transmissibles = European Communicable Disease Bulletin* 29 (13). <https://doi.org/10.2807/1560-7917.ES.2024.29.13.2400123>.

Liu, Wenjun, Paul Pickering, Sebastián Duchêne, Edward C. Holmes, and John G. Aaskov. 2016. "Highly Divergent Dengue Virus Type 2 in Traveler Returning from Borneo to Australia." *Emerging Infectious Diseases* 22 (12): 2146–48.

REVIEWERS' COMMENTS

Reviewer #1 (Remarks to the Author):

Co-Review of revised Nature Communications manuscript NCOMMS-24-21446A

"Genomic surveillance reveals a dengue 2 virus epidemic lineage with a marked decrease in sensitivity to Mosnodenvir"

Comments regarding "Response to Referees Letter/Rebuttal" -----

Authors have a very positive attitude and were very responsive and willing to address reviewer questions/concerns.

Comments regarding revised manuscript -----

In general, the consideration of reviewer comments and subsequent revisions made (inclusion of more samples, sequences, analyses, detailed methods, clinical trial info) have greatly improved the second draft of the manuscript, making it robust and clear, providing a more accurate/objective statement for the need/importance of genomic surveillance for mutations in endemic/epidemic regions where clinical drug use and efficacy may be questioned. Revised manuscript flows nicely, presenting evidence in order to support discussion/conclusion. Ready for publication with few minor/grammatical edits, see below recommendations.

Minor/grammatical suggestions:

Line 17, 342, 346 state "clinical phase 2" (general number format) while a Line 68 states "clinical phase I ... clinical phase II" (roman numeral format). Phase number used to identify trials should be consistent.

Figure 1a – Typo in timeline under Viral Isolation: "Mars 2024" should be "March 2024"

Line 209, 211, 263, 265 – Typos: "91A sequences" should be "V91A sequences"

Figure 1 uses lower case letters (a) (b) (c) while other Figures 2 & 3 use upper case letters (A) (B) (C) . Suggest choosing one format for consistency.

Reviewer #2 (Remarks to the Author):

The manuscript titled "Genomic Surveillance Reveals a Dengue 2 Virus Epidemic Lineage with a Marked Decrease in Sensitivity to Mosnodenvir" has undergone thorough revision based on the feedback provided. The authors have meticulously addressed all the comments and concerns, ensuring that the manuscript now reflects a high standard of scientific rigor and clarity. The revisions have enhanced the overall quality of the work, particularly in terms of data presentation, interpretation of results, and discussion of the implications. The manuscript now presents a compelling and well-supported narrative on the emergence of a dengue 2 virus lineage with reduced sensitivity to Mosnodenvir, which is of significant importance to the field. In its current form, the manuscript is comprehensive, well-organized, and aligns with the journal's standards, making it suitable for publication.

Reviewer #3 (Remarks to the Author):

Reviewer #4 (Remarks to the Author):

The authors have addressed the points I raised in the first review round satisfactorily. However, I have two additional requests as below.

"Finally, we also included DENV-2 sequences from the GISAID database that were not present in GenBank to ensure the comprehensiveness of our results. The corresponding accession numbers are listed in Supplementary Table 6. I could not find Supplementary Table 6 among the files provided in the NPG system. Besides, the authors need to acknowledge the data generators as recommended by GISAID. Please check the guidelines.

Additionally, the authors have made the alignments available through GitHub (https://github.com/Snseli/Dengue_resistance_supplementary/tree/main/datasets). This is interesting for reproducibility, but it must adhere to GISAID's sharing guidelines, which forbid sharing sequences in other repositories. Please review the GISAID guidelines to properly acknowledge the data generators and ensure compliance with GISAID's rules, which protect the data generators' rights over the data."

Gabriel da Luz Wallau

Reviewer #1 (Remarks to the Author):

Co-Review of revised Nature Communications manuscript NCOMMS-24-21446A

“Genomic surveillance reveals a dengue 2 virus epidemic lineage with a marked decrease in sensitivity to Mosnodenvir”

*Comments regarding “Response to Referees Letter/Rebuttal” -----
Authors have a very positive attitude and were very responsive and willing to address reviewer questions/concerns.*

*Comments regarding revised manuscript -----
In general, the consideration of reviewer comments and subsequent revisions made (inclusion of more samples, sequences, analyses, detailed methods, clinical trial info) have greatly improved the second draft of the manuscript, making it robust and clear, providing a more accurate/objective statement for the need/importance of genomic surveillance for mutations in endemic/epidemic regions where clinical drug use and efficacy may be questioned. Revised manuscript flows nicely, presenting evidence in order to support discussion/conclusion.
Ready for publication with few minor/grammatical edits, see below recommendations.*

Answer: We thank the reviewer for this positive assessment of our revised work and for his constructive comments in the previous round of revisions.

Minor/grammatical suggestions:

Line 17, 342, 346 state “clinical phase 2” (general number format) while a Line 68 states “clinical phase I ... clinical phase II” (roman numeral format). Phase number used to identify trials should be consistent.

Figure 1a – Typo in timeline under Viral Isolation: “Mars 2024” should be “March 2024”
Line 209, 211, 263, 265 – Typos: “91A sequences” should be “V91A sequences”
Figure 1 uses lower case letters (a) (b) (c) while other Figures 2 & 3 use upper case letters (A) (B) (C) . Suggest choosing one format for consistency.

Answer: We thank the reviewer for highlighting these errors, they have now been corrected.

Reviewer #2 (Remarks to the Author):

The manuscript titled "Genomic Surveillance Reveals a Dengue 2 Virus Epidemic Lineage with a Marked Decrease in Sensitivity to Mosnodenvir" has undergone thorough revision based on the feedback provided. The authors have meticulously addressed all the comments and concerns, ensuring that the manuscript now reflects a high standard of scientific rigor and clarity. The revisions have enhanced the overall quality of the work, particularly in terms of data presentation, interpretation of results, and discussion of the implications. The manuscript now presents a compelling and well-supported narrative on the emergence of a dengue 2 virus lineage with reduced sensitivity to Mosnodenvir, which is of significant importance to the field. In its current form, the manuscript is comprehensive, well-organized, and aligns with the journal's standards, making it suitable for publication.

Answer: We thank the reviewer for this positive assessment of our revised work and for participating in the reviewing process, it has strongly improved the quality of the manuscript.

Reviewer #3 (Remarks to the Author):

Reviewer #4 (Remarks to the Author):

The authors have addressed the points I raised in the first review round satisfactorily. However, I have two additional requests as below.

"Finally, we also included DENV-2 sequences from the GISAID database that were not present in GenBank to ensure the comprehensiveness of our results. The corresponding accession numbers are listed in Supplementary Table 6. I could not find Supplementary Table 6 among the files provided in the NPG system. Besides, the authors need to acknowledge the data generators as recommended by GISAID. Please check the guidelines.

Answer: We thank the reviewer for this recommendation. We now acknowledge data generators for the GISAID data used in this study using the GISAID doi and table provided by GISAID for the dataset, following GISAID's recommendation.

Additionally, the authors have made the alignments available through GitHub (https://github.com/Snseli/Denque_resistance_supplementary/tree/main/datasets). This is interesting for reproducibility, but it must adhere to GISAID's sharing guidelines, which forbid

sharing sequences in other repositories. Please review the GISAID guidelines to properly acknowledge the data generators and ensure compliance with GISAID's rules, which protect the data generators' rights over the data."

Answer: We thank the reviewer for raising that issue, the sequences from GISAID have now been removed from the Github repository, following GISAID's recommendation.